# Atypical cortical feedback underlies failure to process contextual information in the superior colliculus of *Scn2a*$^{+/-}$ autism model mice

**Leiron Ferrarese** ✉ **& Hiroki Asari** ✉

Atypical sensory integration and contextual learning are common symptoms in autism spectrum disorder (ASD), but how sensory circuits are affected remains elusive. Here we focused on the early visual information processing, and performed in vivo two-photon calcium imaging and pupillometry of mice engaged in an implicit learning task in stable and volatile visual contexts. Wild-type (WT) mice show stimulus-specific contextual modulation of the visual responses in the superior colliculus (SC) and pupil dynamics, whereas *SCN2A*-haploinsufficient ASD-model mice exhibit abnormal modulation patterns. In both genotypes, feedforward inputs from the retina to SC demonstrate no such contextual modulation. In contrast, feedback inputs from the primary visual cortex (V1) show modulation patterns similar to those of SC cells in WT mice, but no modulation in *Scn2a*$^{+/-}$ mice. Furthermore, chemogenetic perturbation reveals that this top-down signaling from V1 to SC mediates the observed contextual modulation both at the neurophysiological and behavioral levels. These results suggest that the corticotectal input is critical for contextual sensory integration in SC, and its anomaly underlies atypical sensory learning in ASD.

Bayesian inference provides a theoretical framework for understanding how the brain combines prior knowledge with incoming sensory information to shape perceptual experiences and guide behavior[1]. In particular, a hierarchical Bayesian learning model has been successfully applied to explain atypical behaviors observed in individuals with autism spectrum disorder[2,3] (ASD): e.g., how overlearning and overestimation of the environmental volatility result in a reduced response to unexpected events in ASD[4]. However, the current experimental evidence supporting this hypothesis is limited in both sample sizes and computational models[3].

How do neuronal circuit functions underlie such differences in learning behavior between individuals with and without ASD? The superior colliculus (SC) is a promising target to address this question.

It is a midbrain hub for sensorimotor transformation implicated in abnormal visual processing in ASD[5,6]. For instance, gaze management and attentional focus are two major functions of SC that are often altered in ASD[7]. SC also plays a role in the initial automatic bias towards global visual processing, which is suggested to be affected in ASD subjects as they typically focus too much on details[6]. Moreover, extensive connections to many other brain regions[8] support that SC interacts with emotional and motor networks, allowing animals to perceive and respond appropriately to sensory stimuli with attentional prioritization, such as the innate looming avoidance responses in mice[9–12]. The involvement of SC in these processes underscores its relevance to the neurophysiological mechanisms of ASD.

Epigenetics and Neurobiology Unit, EMBL Rome, European Molecular Biology Laboratory, Monterotondo, Italy. ✉e-mail: leiron.ferrarese@embl.it; asari@embl.it

Given the convergence of retinal and cortical inputs, SC is expected to be part of a hierarchical recursive system for visual inference in the brain[1,6]. On the one hand, neurons in the superficial layers of SC receive direct visual input from the retina and play a role in the processing of visual motion and orientation[7]. On the other hand, superficial SC neurons also receive a prominent feedback signal from the primary visual cortex[13–15] (V1), which is implicated to help integrate sensory and perceptual information. Recent studies have revealed that the V1-to-SC projection provides contextual gain modulation, enhancing the response amplitude of SC neurons to looming stimuli, while also acting as a negative gain modulator for the saliency of complex visual scenes[16,17]. Furthermore, the corticotectal synapses show nonlinear properties, suggesting their role in regulating sensory-guided innate behavior and modulating attention and learning processes[18,19], although some controversies still remain on the modulatory role of the corticotectal inputs[20–22].

As an increasing number of ASD-associated genes have been identified[23,24], animal models of ASD have been developed to investigate the underlying neurobiological mechanisms. One of the well-established ASD mouse models is the haploinsufficiency for sodium voltage-gated channel alpha subunit 2 (*Scn2a*[+/-] mice)[25,26]. The *SCN2A* gene encodes the Na$_V$1.2 sodium channels that help regulate dendritic excitability of mature cortical pyramidal neurons[27], including those forming the corticotectal pathway[14,26]. Haploinsufficiency of Na$_V$1.2 reduces the backpropagation of action potentials into dendrites, impairing synaptic plasticity and strength in layer 5 (L5) corticofugal pyramidal neurons[26]. Therefore, *Scn2a*[+/-] mice are expected to have atypical corticotectal projections, leading to abnormal neuronal modulations within the SC circuitry.

Here, we developed an implicit visual learning task, similar to the ones used in human studies[4,28,29], but for mice to form probabilistic associations of cue-outcome stimulus sequences in stable and volatile environments. We then combined in vivo two-photon calcium imaging, chemogenetic perturbation, pupillometry, and computational modelling to characterize both neurophysiological and behavioral phenotypes of mice engaged in this task. Over the course of implicit learning, wild-type (WT) mice exhibited a stimulus-specific context-dependent modulation in SC as well as in the corticotectal input from V1 and the pupil dynamics, but not in the input from the retina. A hierarchical Bayesian learning model indicated that a larger fraction of WT SC neurons carries information about the high-level environmental volatility rather than the cue-outcome contingency in itself. In contrast, *Scn2a*[+/-] mice demonstrated anomalous contextual modulation in SC, and no modulation in the V1-to-SC projections, retinal signals, and pupillary responses. Specifically, these SC neurons showed no adaptation to an increased outcome predictability in the stable environment, and largely represented the low-level outcome expectations in a way consistent with the observations in ASD subjects from the perspective of the hierarchical Bayesian learning model[4]. We further found that chemogenetically inhibiting the visual cortical feedback to SC abolished contextual modulations in both WT and *Scn2a*[+/-] mice, both at the physiological and behavioral levels. These results suggest that the corticotectal input plays a critical role in integrating contextual information in SC, and that anomaly in this process underlies atypical sensory learning in ASD.

## Results

### Implicit visual learning paradigm to test behavioral and physiological anomalies in a mouse model of autism spectrum disorder

Individuals with ASD often struggle to learn about probabilistically aberrant events[4]. Understanding the neuronal processes underlying implicit learning and the integration of contextual information will thus provide valuable insights into the neurobiological mechanisms of ASD[3]. In this study, we focused on the early visual system of mice to study (a) how neuronal circuit computation contributes to implicit learning about visual environmental contexts; and (b) how it is altered in a mouse model of ASD (*Scn2a*[+/-]). We thus began by designing an implicit visual learning paradigm based on Bayesian principles[28,29]: i.e., probabilistic associative learning of cue-outcome stimulus sequences, much as those experimental paradigms used in human studies[4].

Our implicit learning task was carefully tailored with visual stimuli to elicit robust responses in the mouse SC[7], such as static gratings with different orientations[30,31] and looming and sweeping stimuli that trigger SC-mediated arousal and innate defensive behavior[9–12]. Specifically, we used vertical and horizontal gratings as neutral cues; and a combination of sweeping and looming dark spots as aversive outcomes while a gray screen (omitted stimulus) as a non-aversive outcome (Fig. 1a). Since the task did not involve any reward or punishment, we exploited innate visual threats in the outcome stimulus to motivate animals to recognize stimulus sequences. To increase unpredictability, we introduced uncertainty by randomizing the type and direction of the visual threat at each presentation (Fig. 1a; see Methods for details).

The task followed a probabilistic learning scheme modeled by a hierarchical Gaussian filter[4,28,29] (HGF), consisting of a stable

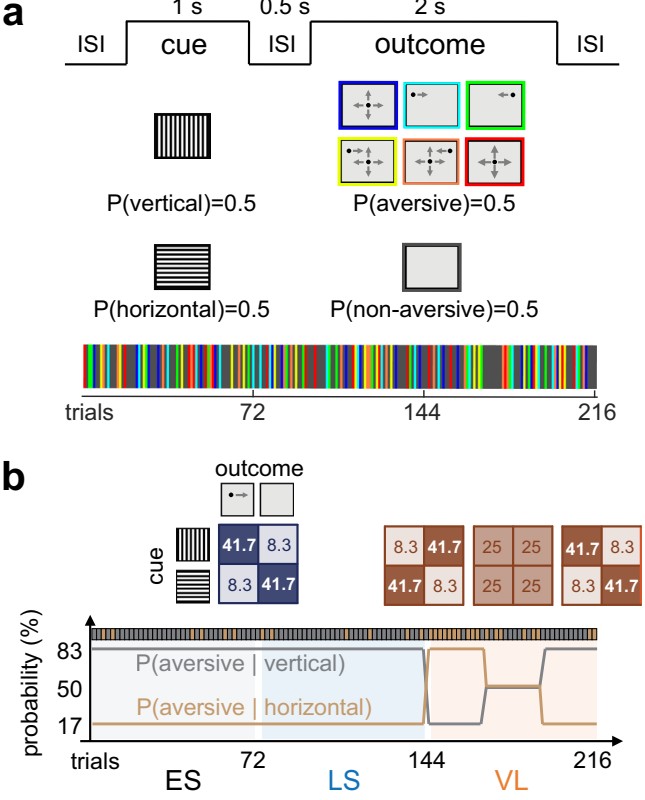

**Fig. 1 | Implicit visual learning task to study adaptive sensory processing in mice. a** Schematic of the implicit visual learning task. In each trial, a neutral cue (vertical or horizontal gratings; 1 s) is followed by an aversive outcome (looming and/or sweeping dark spot, randomly chosen from six different types; 2 s) or non-aversive one (omitted; 2 s) with an inter-stimulus interval (ISI) of 0.5 s. In total, the task consisted of 216 trials, including 108 aversive outcome trials (color-coded) and 108 non-aversive ones (dark gray). **b** The conditional probability of cue-outcome contingencies (expected, gray; unexpected, light brown) during stable and volatile environments (joint probability matrix shown on top). The stable environment lasts twice as long as the volatile one (VL, orange), and is divided into early-stable (ES) and late-stable (LS) epochs for analysis (72 trials each; light gray and light blue, respectively). Animals are expected to form cue-outcome associations during the stable environment, while such associations are violated in the subsequent volatile environment.

environment followed by a volatile one (Fig. 1b). During the stable environment (144 trials in total), the aversive outcome stimuli were primarily preceded by one type of neutral cue stimuli (vertical gratings; 83%). Mice were then expected to form probabilistic associations of these cue-outcome stimulus sequences (cue-outcome contingencies). At a small probability, however, the aversive stimuli followed the other type of cue stimuli (horizontal gratings; 17%). This allows animals to determine whether an unexpected outcome indicates a deviation from a stable stimulus pattern or a change in the pattern itself. Ideal observers would consider the latter to be less likely in the late-stable epoch (LS; last 72 trials) than in the early-stable epoch (ES; initial 72 trials). However, those with ASD traits, such as reduced adaptability[32–34], would not, thereby affecting the responses in the LS epoch. In the following volatile epoch (VL, 72 trials in total), we then introduced variations in the probability of presenting aversive stimuli after each neutral stimulus to challenge the animal's expectations on the stimulus sequences. Animals with hypersensitivity[35], for example, would then reveal anomalies in the VL epoch. Different patterns of anomalies should thus be observed in our task, depending on the ASD-associated phenotypes that the subject animal has.

### Wild-type and $Scn2a^{+/-}$ mice exhibit distinct contextual modulation patterns in the visual responses of the superior colliculus

We began with neurophysiological assays on awake, head-fixed mice expressing genetically encoded calcium indicators (GECIs) in SC somata delivered via adeno-associated viral vectors (AAVs; see Methods for details; Fig. 2a, b). Specifically, we exposed the mice to the implicit visual learning task as described above, and simultaneously performed in vivo two-photon calcium imaging of the superficial layer of the medial-posterior SC area through a cranial window, together with pupillometry for behavioral monitoring[36,37] (Supplementary Fig. 1a).

On average we recorded 140 ± 39 cells per animal (mean ± standard deviation; 17 WT and 16 $Scn2a^{+/-}$ mice) and estimated their spiking activity using CaImAn[38] ($Z(t)$ in z-score, see Methods for details). As expected[7,30,31], superficial SC neurons generally responded well to the onset and/or offset of the static gratings presented as cue stimuli (Supplementary Figs. 1 and 2). Here, we considered cells as responsive if the trial average of their visual response dynamics exceeded a predetermined threshold ($\max[z(t)] > 1$, where $z(t) = \langle Z(t) \rangle_{trials}$; see Methods for details). We then identified orientation-selective responses in about a quarter of the cells[22,30,31,39] (WT, 23 ± 10%; $Scn2a^{+/-}$, 22 ± 9%), with a larger fraction preferring vertical orientation[40,41] (Supplementary Table 1). Such response bias, however, varied across individual animals (vertical bias, 8 WT and 6 $Scn2a^{+/-}$ mice; horizontal bias, 4 WT and 2 $Scn2a^{+/-}$ mice; $p < 0.05$, $\chi^2$-test on data from each animal with Bonferroni correction), and no significant difference was observed between WT and $Scn2a^{+/-}$ animals ($p = 0.31$, Kolmogorov-Smirnov test; see Methods for details).

Of those recorded cells, about half were responsive to the looming and/or sweeping stimuli presented as aversive outcome (74 ± 28 cells per recording), whereas in most cells we found virtually no response to the omitted stimulus presented as non-aversive outcome (gray screen; Supplementary Fig. 3). Due to retinotopy in the early visual system, (a) each aversive stimulus drove cells at different timings in our experimental design; and (b) each cell showed distinct response patterns to different aversive stimuli. Furthermore, the low number of repetitions for each aversive stimulus makes it difficult to reliably assess neuronal responses at the single-cell level, especially due to eye movements. We thus combined all the observed responses across population to gain enough statistical power in the subsequent analyses.

To examine the SC dynamics during the implicit learning task, we compared the aversive stimulus responses at the population level across the three analysis epochs in our paradigm (ES, LS, and VL; Fig. 2c–e). For each responsive cell, we calculated the average response strength during the stimulus presentation period in each epoch

($\bar{z} = \langle z(t) \rangle_t$; e.g., Fig. 2c), and subtracted the mean across the epochs for normalization ($\Delta z = \bar{z} - \langle \bar{z} \rangle_{epochs}$; e.g., Fig. 2e). Using this normalized response, $\Delta z$, we found that SC neurons in WT animals responded significantly weaker to the aversive stimuli during the LS epoch than the ES or VL epoch (Fig. 2e). Such changes in the population response strength were relatively stronger for later components of the aversive stimulus responses than for the early response components (Supplementary Fig. 4), while not observed for the cue stimulus responses (Supplementary Fig. 2). These suggest that the responses of SC neurons in WT mice are modulated by the visual contexts in two ways. First, when the stimulus pattern is reliably predictable during the stable environment, the responses to the aversive stimuli become selectively and progressively attenuated over trials (adaptation). Second, when the stimulus pattern deviates from an animal's expectation during the volatile environment, the response of SC neurons is reset to the baseline and becomes as high as in the initial ES epoch before adaptation.

In contrast, $Scn2a^{+/-}$ mice displayed different context-dependent modulations in SC during the implicit visual learning task: no adaptation in the stable environment, but increased neuronal responses to the aversive stimuli and decreased responses to the cue stimuli during the volatile environment (Fig. 2f–h and Supplementary Fig. 2). Changes in the population response strength were relatively larger for the early response components than for the later ones (Supplementary Fig. 4). These response patterns of SC neurons in $Scn2a^{+/-}$ mice were significantly different from those observed in WT ($p$(epoch × group) = 0.0007, 3-way nested analysis of variance (ANOVA) on $\bar{z}$; see Methods for details). This suggests that, unlike WT animals, SC neurons in $Scn2a^{+/-}$ mice fail to selectively attenuate their aversive stimulus responses even when the stimulus pattern is kept predictable. They are, however, still sensitive to increased volatility, and thus their response during the VL epoch diverged from the baseline activity in a stimulus-specific manner. This is consistent with the observations on ASD subjects who tend to overestimate environmental volatility in stable environments, and overreact to increased unpredictability in volatile environments[4].

### Context-dependent modulation is absent in retinal output responses

How do SC neurons achieve such stimulus-specific context-dependent modulations? One possibility is that the contextual modulation in SC is inherited from retinal ganglion cells (RGCs). To investigate the contribution of bottom-up RGC input to SC, we performed in vivo two-photon calcium imaging of RGC axon terminals in SC[37] while the animals were engaged in the implicit visual learning task (Fig. 3a). On average we recorded 323 ± 94 putative axonal boutons per animal (mean ± standard deviation; 5 WT and 5 $Scn2a^{+/-}$ mice), of which 208 ± 87 were responsive to the looming and/or sweeping stimuli ($\max[z(t)] > 0.75$). In both WT and $Scn2a^{+/-}$ mice, RGC axon terminals in SC showed consistent responses to the aversive stimuli regardless of the visual context (Fig. 3b, e), and we found no significant contextual modulation across ES, LS, and VL epochs at the population level (Fig. 3c, f). No contextual modulation was observed in the population responses to the omitted or cue stimuli, either (Supplementary Figs. 2–4). This excludes the possibility that the contextual modulation in SC arises from the retina.

### Corticotectal inputs show similar context-dependent modulation patterns as local neurons in the superior colliculus of wild-type mice, but no modulation in $Scn2a^{+/-}$ mice

Another possible source for the contextual modulation in SC is the corticotectal projection. Besides sensory input from RGCs, the superficial layer of SC receives substantial feedback signals from L5 pyramidal neurons in V1[14]. These V1 L5 neurons integrate feedforward visual information through their basal dendrites and feedback

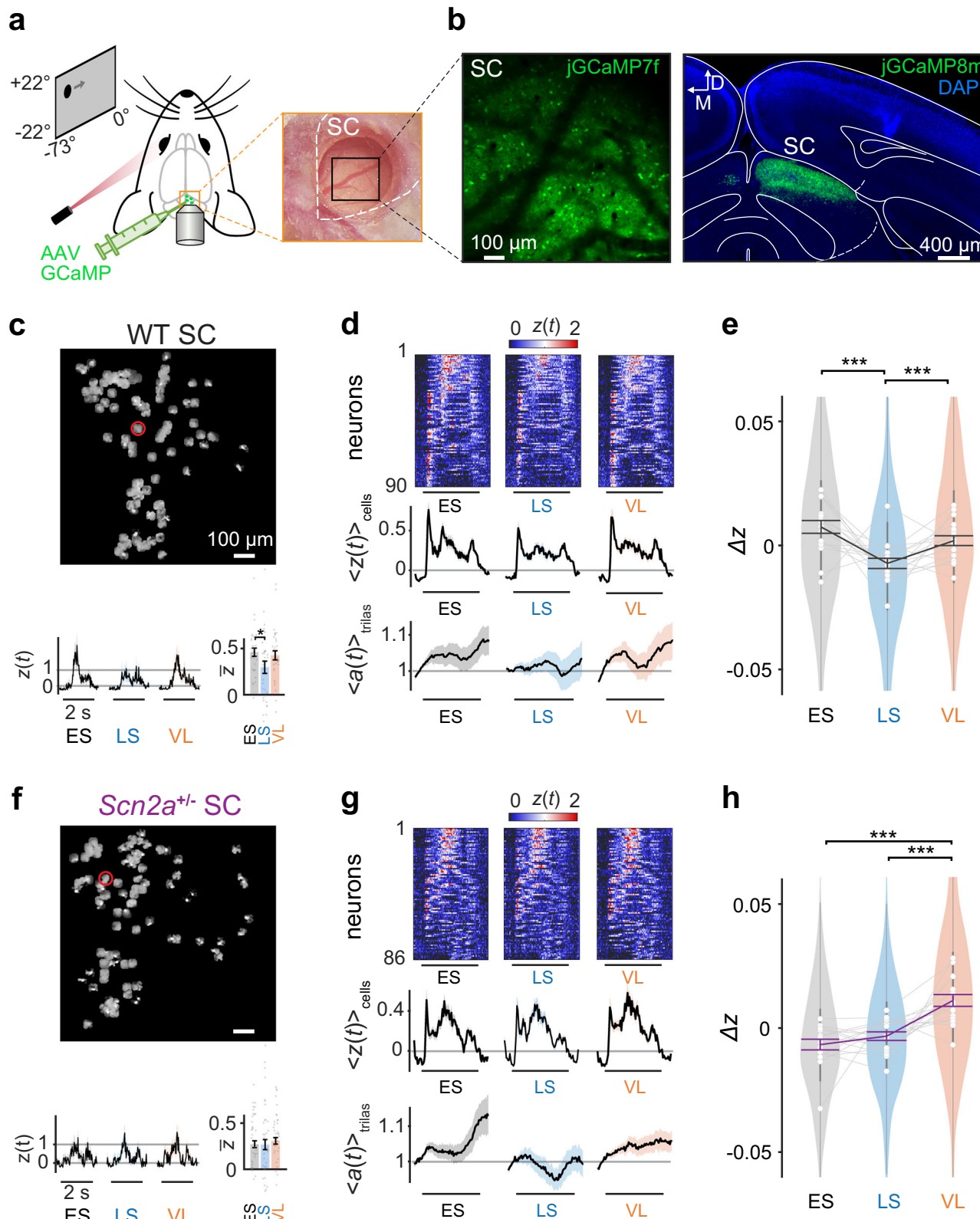

contextual information via the apical dendrites[42], and provide output to SC for controlling the response gain[16–19]. Cortical pyramidal neurons in *Scn2a+/−* mice have impaired synaptic functions due to a reduced expression of the Na$_V$1.2 sodium channel[26]. This may result in a failure to incorporate visual and contextual information in V1, which in turn may lead to an anomalous modulation of SC as observed in our paradigm (Fig. 2 and Supplementary Figs. 1–4).

To examine the role of V1 feedback in the contextual modulation of SC, we first performed in vivo two-photon calcium imaging of the axon terminals of V1 projection neurons in SC while animals were engaged in the implicit visual learning task (Fig. 4a). On average, we recorded $412 \pm 104$ putative axonal boutons per animal (mean ± standard deviation; 8 WT and 7 *Scn2a+/−* mice), with $167 \pm 51$ boutons responsive to the looming and/or sweeping stimuli ($\max[z(t)] > 0.75$).

**Fig. 2 | Neurons in the superior colliculus show contextual adaptation to a stable environment in wild-type mice but not in *Scn2a*[+/−] mice. a** Schematic diagram of the experimental setup for in vivo two-photon calcium imaging of the superior colliculus (SC, outlined with white dashed line; black square, field of view). **b** Representative two-photon image frame and post hoc histology image (coronal slice; blue, 4′,6-diamidino-2-phenylindole (DAPI) stain), demonstrating a high GCaMP expression (green) in the right SC of a wild-type (WT) mouse. **c** Responses of a representative WT SC neuron (red circle on the spatial map of the simultaneously recorded cells) to aversive stimuli (horizontal bar, 2 s) in each analysis epoch (ES, early stable; LS, late stable; VL, volatile): time series, mean and the standard error of the mean (s.e.m., shaded area) over 36 aversive outcome trials, $z(t) = \langle Z(t) \rangle_{\text{trials}}$; bar graph, mean and s.e.m. over the stimulus period, $\bar{z} = \langle z(t) \rangle_t$; *, $p < 0.05$ with two-sided Mann-Whitney $U$-test. **d** Heatmap $z(t)$ and the population average $\langle z(t) \rangle_{\text{cells}}$ (shaded area, s.e.m.) of the aversive stimulus responses of WT SC

neurons in each epoch (90 cells from **c**; sorted with max[$z(t)$] > 1). Corresponding pupil dynamics are shown at the bottom ($\langle a(t) \rangle_{\text{trials}}$; shaded area, s.e.m.). **e** Distributions of the normalized WT SC responses across epochs ($\Delta z = \bar{z} - \langle \bar{z} \rangle_{\text{epochs}}$) in violin plots (1252 cells from 17 mice): gray lines with open circles, average for each mouse; vertical bars, interquartile range; thick line graph with error bars, mean ± s.e.m.; ***, $p < 0.001$ with two-sided Tukey-Kramer test for pairwise comparisons. See Methods and Supplementary Table 1 for details. Corresponding data for *Scn2a*[+/−] mice: (**f**), a representative cells' location (red circle) and responses in each epoch: time series, mean and s.e.m. over trials; bar graphs, mean and s.e.m. over the stimulus period; (**g**), a representative animal's data (86 cells across 36 trials; mean and s.e.m.); (**h**), population data in violin plots (1201 cells in total from 16 *Scn2a*[+/−] mice); gray lines with open circles, average for each mouse; vertical bars, interquartile range; thick line graph with error bars, mean ± s.e.m.; ***, $p < 0.001$ with two-sided Tukey-Kramer test for pairwise comparisons.

In WT mice, the aversive stimulus responses of the V1 axon terminals in SC showed significant context-dependent modulation (Fig. 4b, c), especially for later response components (Supplementary Fig. 4), resembling the dynamics observed in SC somata. The responses were attenuated during the LS epoch compared to the ES epoch, consistent with adaptation during stable environments, and returned to the baseline levels during the VL epoch, reflecting the sensitivity to increased volatility. Unlike SC cells, however, these V1 axons showed a significant reduction in the cue stimulus responses during the VL epoch (Supplementary Fig. 2), and modest responses to the omitted stimulus, demonstrating adaptation during stable environments but no recovery in the volatile environment (Supplementary Fig. 3). This highlights the stimulus specificity of the contextual modulation and distinct information processing in different areas of the early visual system.

In contrast, V1-to-SC projections in *Scn2a*[+/−] mice displayed no context-dependent modulation (Fig. 4d–f). Unlike SC cells, these V1 axonal boutons failed to adapt to stable environments or sensitize in volatile environments, and kept stable response levels throughout the learning task regardless of the stimuli (Supplementary Figs. 2–4). Haploinsufficiency of Na$_V$1.2 impairs synaptic plasticity and dendritic excitability in L5 corticofugal neurons[14,26,27], including those projecting from V1 to SC. Thus, the observed anomalies of the V1-to-SC projections in *Scn2a*[+/−] mice may arise from the direct cell-intrinsic effects of the *SCN2A* mutation on those corticotectal neurons, or indirect neuronal circuit effects on the upstream visual cortical processing.

### Chemogenetically inhibiting corticotectal projection abolishes context-dependent modulation in the superior colliculus

To further examine how V1 inputs affect the visual response of SC, we next employed a chemogenetic approach to selectively inhibit the V1 L5 corticotectal neurons using a designer receptor exclusively activated by designer drug[43] (DREADD). Specifically, we achieved selective expression of modified human M4 muscarinic acetylcholine receptors for inhibitory DREADD (hM4Di) in the target neuronal population by injecting Cre-dependent AAV8 carrying the hM4Di receptor into V1 and AAV2retro carrying Cre-recombinase into SC (Fig. 5a). Histological analyses confirmed that axonal projections were clearly visible in the ipsilateral SC of all mice examined (e.g., Fig. 5b; 7 mice each for WT and *Scn2a*[+/−]); and that these projections originated mostly from the V1 L5 area, where labelled cells occupied 31 ± 22% (mean ± standard deviation), mostly in the medial-anterior part. Systemic administration of the DREADD actuator, clozapine N-oxide (CNO), then activated hM4Di, hence effectively inhibited the V1 neurons projecting to SC. As before, AAVs carrying GECIs were also injected into SC for monitoring the visual responses of SC neurons with in vivo two-photon microscopy. Co-injection of AAVs carrying GECIs and those carrying Cre-recombinase into SC ensures that the recorded SC cells were surrounded by the silenced V1 axons, thereby maximizing the chance to

observe the effects of the chemogenetic manipulations. However, experiments with and without CNO injection were conducted in different sessions. Therefore, the recorded SC populations were not necessarily the same, even from the same animal.

Chemogenetic inhibition of the projection from V1 to SC abolished the context-dependent modulation of SC responses to the aversive stimuli in both WT and *Scn2a*[+/−] mice (Fig. 5c–f). After the intraperitoneal injection of CNO, SC neurons did not show adaptation during the LS epoch or recovery/enhancement during the VL epoch (e.g., Fig. 5c, e). As a result, the overall response level of SC cells did not significantly change throughout the implicit visual learning task (Fig. 5d, f), though the early response components of WT mice increased in the VL epoch, resembling the dynamics of intact *Scn2a*[+/−] animals (Supplementary Fig. 4), and the cue stimulus responses showed a tendency to decrease over trials after blocking V1 (Supplementary Fig. 2). Such response patterns to the aversive stimuli across epochs were significantly different from those in intact animals ($p$(epoch × group) = 0.0005, 3-way nested ANOVA; see Methods for details). Importantly, hM4Di expression in V1 or CNO administration alone had no effects on the contextual modulation of SC cells in WT mice (Supplementary Fig. 5). We further confirmed via receptive field analyses that the basic visual response properties of SC cells, RGC axons and V1 axons in SC were overall similar between WT and *Scn2a*[+/−] mice (Supplementary Fig. 6). Taken together, we suggest that feedback input from V1 plays a key role in modulating the aversive stimulus response of SC neurons in a context-dependent manner, and that anomalies of the V1-to-SC signals likely underlie the impairments of the context-dependent visual processing in the SC of *Scn2a*[+/−] mice.

### Pupil dynamics reflect context-dependent stimulus pattern recognition in wild-type but not in *Scn2a*[+/−] mice

Together with neuronal activities, we collected pupillometric data (Fig. 6) as a proxy to monitor cognitive processes of the animals[44]. Here, we in particular interpreted pupil dynamics as a reflection of an animal's stimulus pattern recognition[45–47]. Indeed, all mice exhibited stereotypical pupil dynamics during the implicit visual learning task: i.e., pupil constriction in response to the cue stimuli, and pupil dilation during the outcomes, especially when the aversive stimuli were presented (Supplementary Figs. 1–3). This consistent response over time across epochs suggests that they were attentive throughout the experiments.

In WT animals, we found a smaller magnitude of pupil dilation upon aversive stimulus presentations during the volatile environment than the stable environment (Fig. 6d, h), while no substantial change in the pupil dynamics was observed during the cue or omitted stimulus presentations (Supplementary Figs. 2 and 3). Such stimulus-specific context-dependent modulation of the pupil dynamics was not observed when the feedback signal from V1 to SC was chemogenetically blocked (Fig. 6f, j), and the resulting modulation pattern was significantly different from that of the negative controls

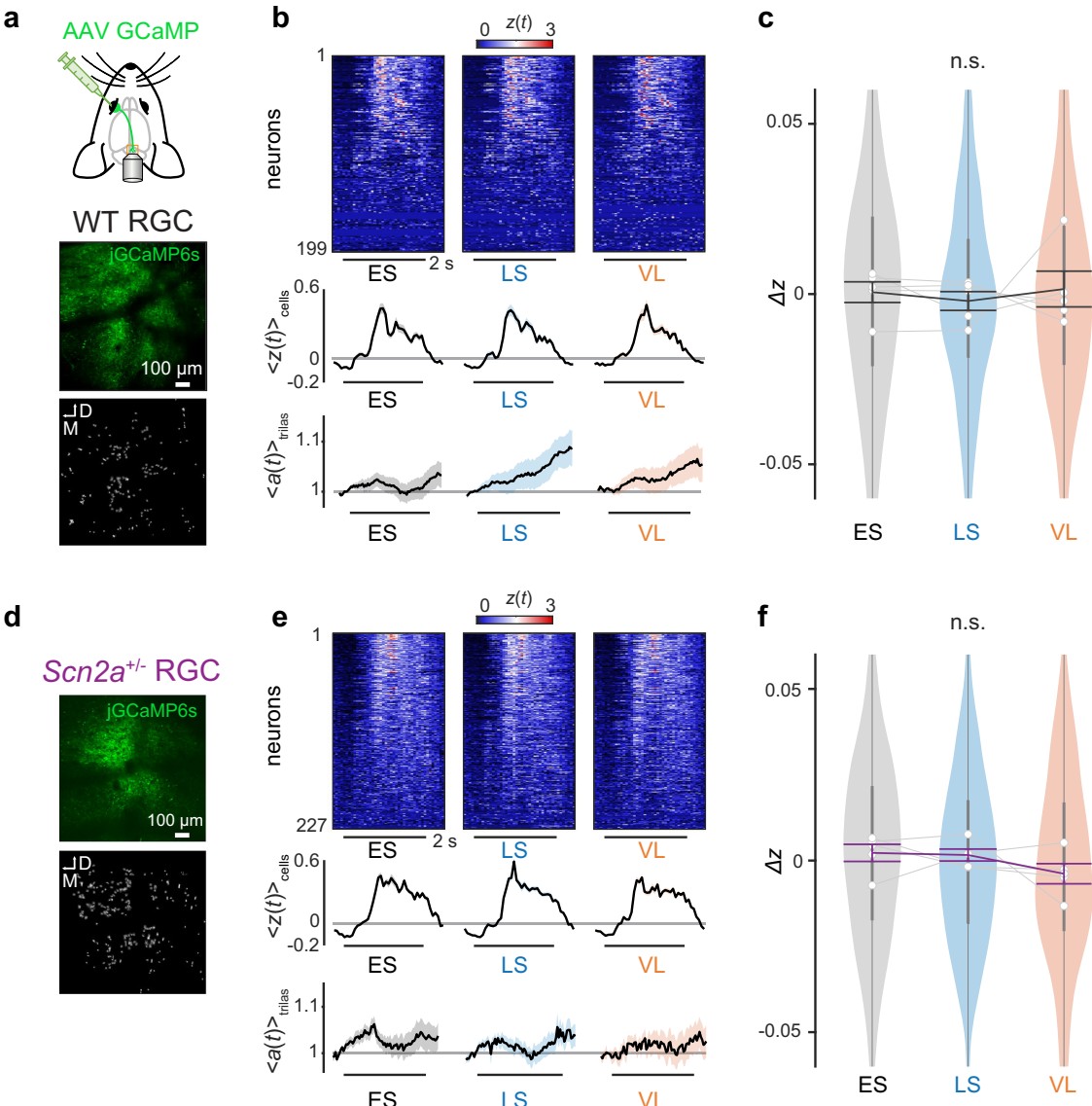

**Fig. 3 | Axon terminals of retinal ganglion cells in the superior colliculus do not show any context-dependent modulation during the implicit visual learning task. a** Schematic diagram of the experimental setup for in vivo two-photon calcium imaging of retinal ganglion cell (RGC) axon terminals in the superior colliculus (SC). An example image frame and the extracted spatial map of RGC boutons are shown at the bottom. **b** Aversive stimulus responses of simultaneously recorded RGC boutons (heatmap of $z(t)$, sorted by $\max[z(t)]$), the population average ($\langle z(t) \rangle_{cells}$; 199 boutons in total from (**a**); shaded area, s.e.m.), and pupil size dynamics ($\langle a(t) \rangle_{trials}$; shaded area, s.e.m.) from a representative wild-type (WT) mouse in each analysis epoch (ES early stable, LS late stable, VL volatile); horizontal bar, stimulation period (2 s). **c** Distributions of the normalized RGC population responses across epochs ($\Delta z$) in violin plots (877 boutons from 5 WT mice): gray lines with open circles, average for each mouse; vertical bars, interquartile range; thick line graph with error bars, mean ± s.e.m. No significant difference was found by pairwise comparisons (two-sided Tukey–Kramer test). Corresponding data for $Scn2a^{+/-}$ mice: (**d**), example image frame of simultaneously recorded RGC axons and putative bouton locations; (**e**), representative data of simultaneously recorded responses of RGC axons to the aversive stimuli (horizontal bar, 2 s; average across 227 boutons and s.e.m.); (**f**), population data in violin plots (1208 responsive boutons in total from 5 animals): gray lines with open circles, average for each mouse; vertical bars, interquartile range; thick line graph with error bars, mean ± s.e.m. No significant difference was found by pairwise comparisons (two-sided Tukey–Kramer test).

($p$(epoch × group) = 0.01, 2-way ANOVA between animals expressing hM4Di with and without CNO, Supplementary Fig. 5c; $p$(epoch × group) = 0.04, 2-way ANOVA between CNO-administered animals with and without hM4Di expression, Supplementary Fig. 5f).

In contrast, $Scn2a^{+/-}$ animals generally showed a smaller pupil size than WT animals throughout the task (24 $Scn2a^{+/-}$ mice, 0.62 (0.44) mm², median (interquartile range) of the median pupil area; 25 WT mice, 0.79 (0.28) mm²; $p$ = 0.006, Mann–Whitney U-test; Fig. 6a–c),

and displayed no context-dependent modulation (Fig. 6e, i and Supplementary Figs. 2 and 3). Chemogenetic inhibition of the corticotectal projection did not affect the pupil dynamics in $Scn2a^{+/-}$ mice (Fig. 6g, k). The modulation pattern of the pupil dynamics across epochs was significantly different between WT and the other three experimental groups ($p$(epoch × group) = 0.04; 2-way ANOVA; see Methods for details). This supports that the influence of the corticotectal signals extends beyond the neuronal activity in SC, likely playing a role in the

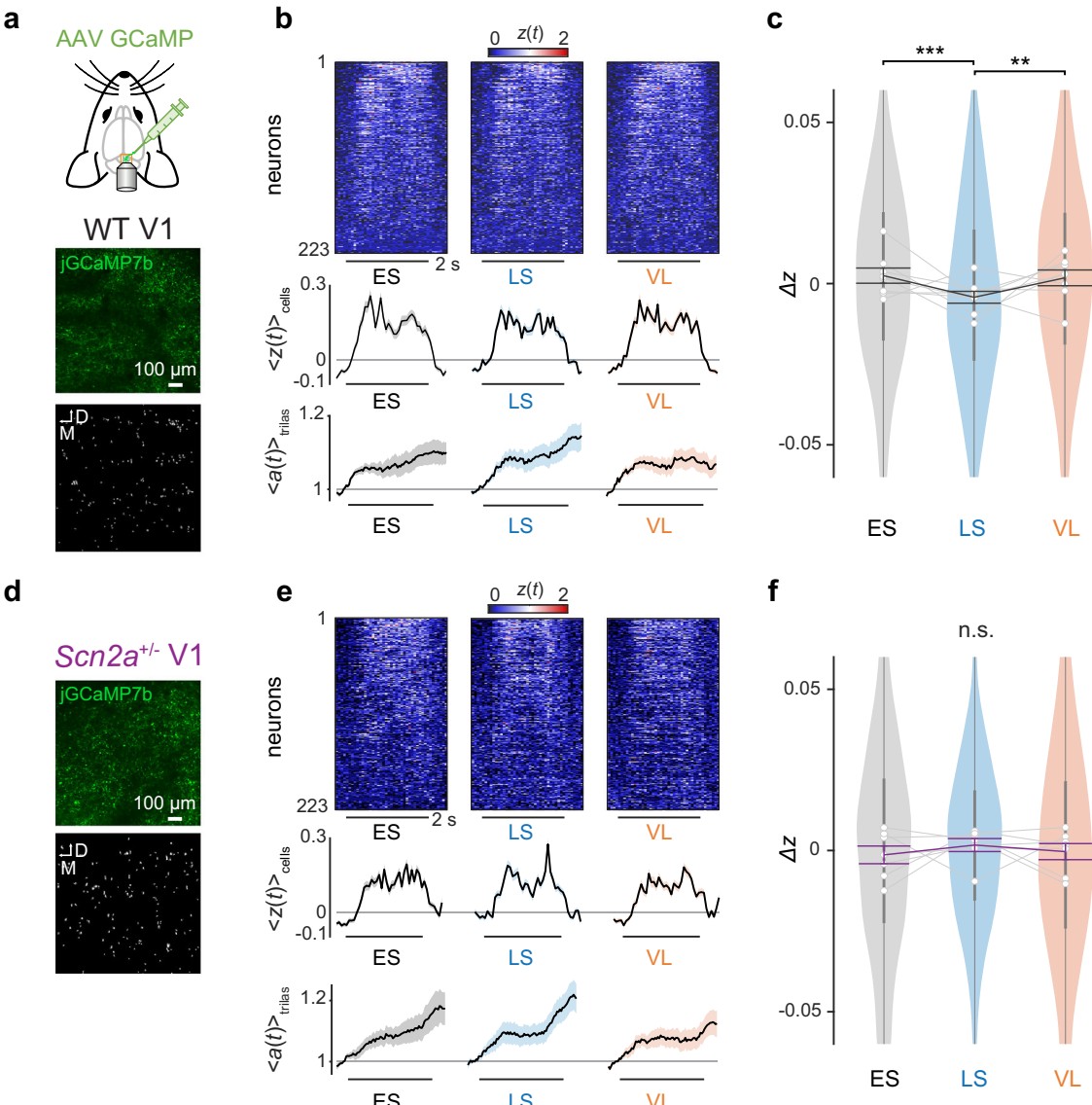

**Fig. 4 | Axon terminals of primary visual cortical cells in the superior colliculus show similar context-dependent modulation to the collicular neurons in wild-type mice, but not in *Scn2a*⁺/⁻ animals. a** Schematic diagram of the experimental setup. Animals were injected with adeno-associated viruses (AAVs) carrying GCaMP7b into the primary visual cortex (V1) and underwent cranial window surgery over the superior colliculus (SC) to monitor the axonal activity of V1 cells in vivo. An example image frame and the extracted spatial map of V1 axon terminals in SC are shown at the bottom. **b** Aversive stimulus responses of simultaneously recorded boutons of V1 axons (heatmap of $z(t)$, sorted by max[$z(t)$]), the population average ($\langle z(t) \rangle_{cells}$; 223 boutons in total from (**a**); shaded area, s.e.m.) and pupil size dynamics ($\langle a(t) \rangle_{trials}$; shaded area, s.e.m.) from a representative WT mouse in each analysis epoch (ES early stable, LS late stable, VL volatile); horizontal bar, stimulation period (2 s). **c** Distributions of the normalized axonal responses across epochs

($\Delta z$) in violin plots (1376 boutons from 8 WT mice): gray lines with open circles, average for each mouse; vertical bars, interquartile range; thick line graph with error bars, mean ± s.e.m.; ***, $p < 0.001$; **, $p < 0.01$ with two-sided Tukey–Kramer test for pairwise comparisons. Corresponding data for *Scn2a*⁺/⁻ mice: (**d**), example image frame of simultaneously recorded V1 axons projected to SC and putative bouton locations; (**e**), representative data of simultaneously recorded responses of V1 axons to the aversive stimuli (horizontal bar, 2 s; average across 223 putative boutons and s.e.m.); (**f**), population data in violin plots (1254 responsive boutons in total from 7 mice): gray lines with open circles, average for each mouse; vertical bars, interquartile range; thick line graph with error bars, mean ± s.e.m. No significant difference was found by pairwise comparisons (two-sided Tukey–Kramer test).

contextualization of the visual scene even at the perceptual level; and that *Scn2a*⁺/⁻ mice have altered V1-to-SC signaling, resulting in a failure to properly recognize the stimulus patterns during the implicit learning task.

Taken together, we found that the modulation of the pupil dynamics (Fig. 6) was well linked to that of the neuronal dynamics in SC and V1-to-SC projection (Figs. 2–5): i.e., the greater the pupil dilation

was, the weaker the responses of SC cells and V1 input were in our experimental paradigm, much as was demonstrated in previous studies[48,49]. Most importantly, *Scn2a*⁺/⁻ mice showed impaired neurophysiological and behavioral phenotypes, both due to anomalous signaling from V1 to SC. This highlights a crucial role of the top-down corticotectal pathway in forming perceptual associations to recognize and properly react to visual contexts[18,19], and a possible relationship

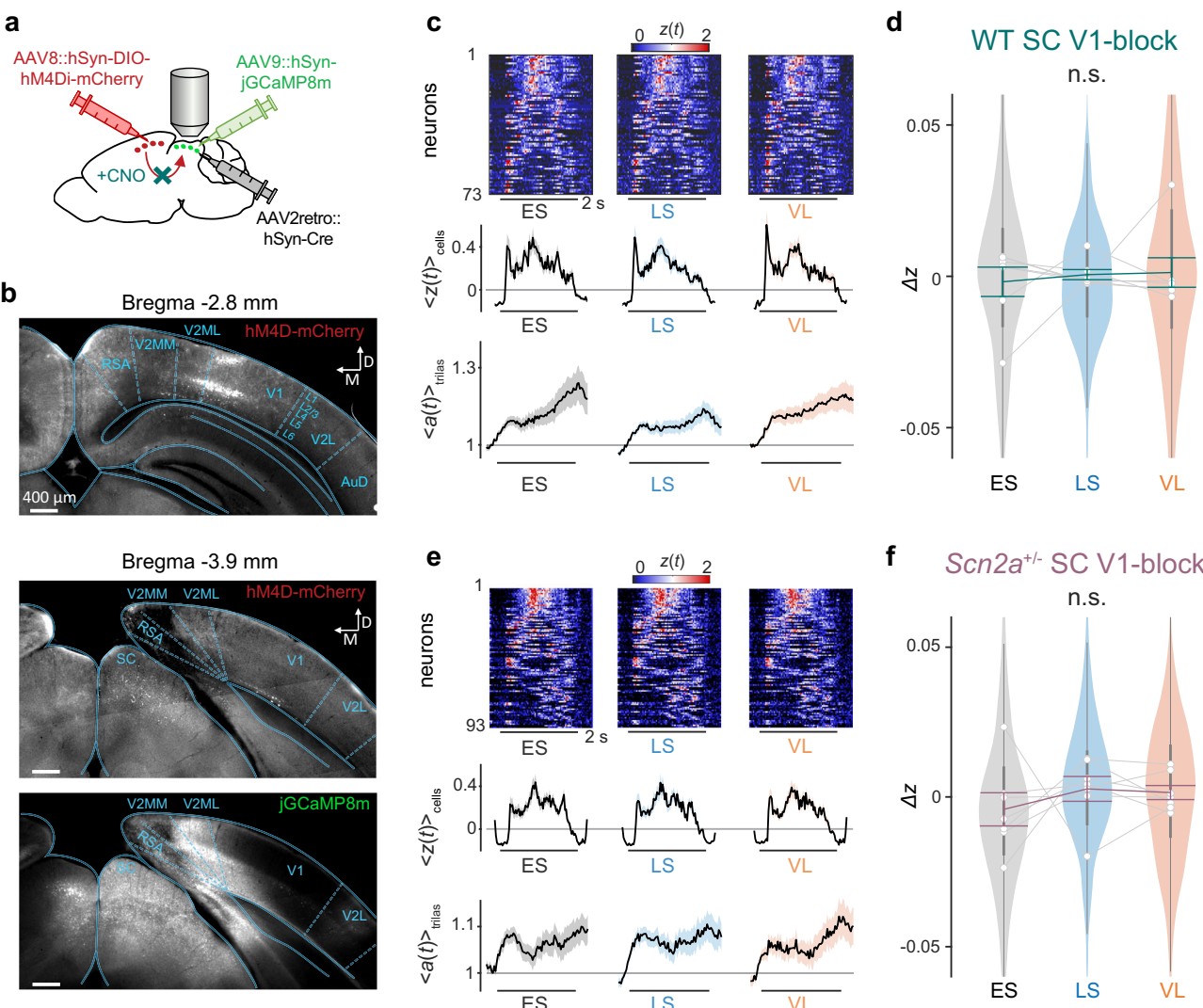

**Fig. 5 | Inhibition of the corticotectal projection abolishes context-dependent modulation in the superior colliculus. a** Schematic of the viral vector strategy for chemogenetic inhibition of the projection from the primary visual cortex (V1) to the superior colliculus (SC). AAV8::hSyn-DIO-hM4Di-mCherry was injected into V1, while AAV2retro::hSyn-Cre and AAV9::hSyn-jGCaMP8m were co-injected into SC. Systemic administration of clozapine N-oxide (CNO) selectively inhibited the V1-to-SC projection by activating the hM4Di receptors. **b** Histological verification of the hM4Di receptor expression (with mCherry) in the V1 layer 5 (L5) cells (top; Bregma −2.8 mm) and their axons in SC (middle; Bregma −3.9 mm). Expression of hM4Di receptors was also observed in the V1 layer 2/3 (L2/3) cells, likely because they project to the secondary visual cortical area (V2MM, mediomedial; V2ML, mediolateral), where we had a leakage of the AAV2retro virus during injection, as indicated by jGCaMP8m signals via the co-injected AAV9 virus (bottom). **c** Aversive stimulus responses of SC neurons (heatmap of $z(t)$, sorted by max[$z(t)$]), the population average ($\langle z(t) \rangle_{cells}$ over 73 cells; shaded area, s.e.m.), and pupil size dynamics ($\langle a(t) \rangle_{trials}$; shaded area, s.e.m.) in each epoch (ES early stable, LS late stable, VL volatile, horizontal bar, stimulation period), recorded from a representative wild-type (WT) mouse after chemogenetically blocking V1-to-SC projection. **d** Distributions of the normalized WT SC responses across epochs ($\Delta z$) after chemogenetically blocking V1-to-SC projection in violin plots (423 cells from 7 WT mice); gray lines with open circles, average for each mouse; vertical bars, interquartile range; thick line graph with error bars, mean ± s.e.m. No significant difference was found by pairwise comparisons (two-sided Tukey−Kramer test). Corresponding data for $Scn2a^{+/-}$ mice with chemogenetic inhibition of V1-to-SC projections: (**e**), representative data of simultaneously recorded responses of SC neurons to the aversive stimuli (horizontal bar, 2 s; average across 93 cells and s.e.m.); (**f**), population data in violin plots (550 cells in total from 7 mice): gray lines with open circles, average for each mouse; vertical bars, interquartile range; thick line graph with error bars, mean ± s.e.m. No significant difference was found by pairwise comparisons (two-sided Tukey−Kramer test).

between ASD and dysfunctions of V1 L5 projection neurons and their upstream circuits[5,6,50].

### Corticotectal signals in $Scn2a^{+/-}$ mice fail to convey visual contextual information from the hierarchical Bayesian inference perspective

Thus far, we made population-level analyses on our neurophysiological and behavioral data during the implicit visual learning task, and identified that (1) V1 provides critical modulatory signals to SC; and (2) anomalies of such corticotectal signals likely underlie the absence of stimulus-specific context-dependent modulation in the SC and pupil

dynamics of $Scn2a^{+/-}$ mice (Figs. 2–6). What do those individual SC neurons and inputs to SC represent? To address this question, we simulated how an ideal Bayesian observer should respond to the sequence of the cue-outcome contingencies in our experimental paradigm[1,28], and compared the dynamics of the underlying model parameters with those of the observed neuronal responses on a trial-to-trial basis to gain insights from the Bayesian inference perspective (Fig. 7).

Among many Bayesian inference models[51], here we employed an HGF model, similar to the one used to explain the differences between subjects with and without ASD in their behavioral responses to

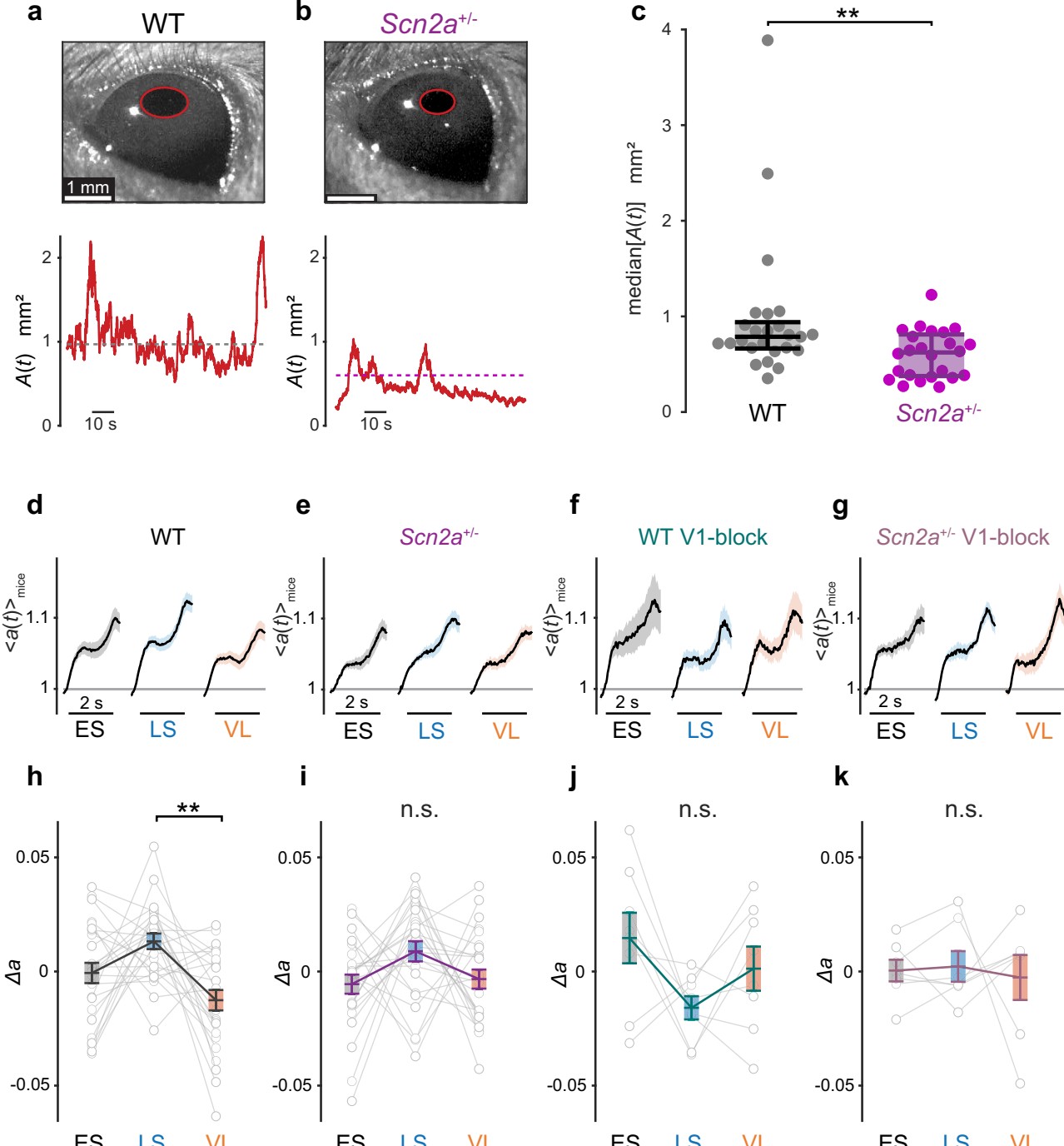

**Fig. 6 | Volatility-dependent pupillometric response patterns in wild-type but not in *Scn2a*⁺/⁻ mice.** Representative image frame of the left eye (red, pupil outline) of wild-type (WT, **a**) and *Scn2a*⁺/⁻ mice (**b**), and the pupil area dynamics $A(t)$ in mm² during visual stimulation (dashed line, median), respectively. See also Supplementary Fig. 1 for example data traces. **c** Median pupil size of 25 WT and 24 *Scn2a*⁺/⁻ mice during the implicit visual task (error bar, median and interquartile range): **, $p < 0.01$, two-sided Mann–Whitney $U$-test. **d–k** Pupil size dynamics (**d–g**, average proportional change in pupil area, $\langle a(t)\rangle_{mice}$; shaded area, s.e.m.) and normalized change in pupil size (**h–k**, $\Delta a = \bar{a} - \langle \bar{a}\rangle_{epochs}$; thin lines, individual animals; error bars, population average and s.e.m.) in response to the aversive stimulus (horizontal bar, 2 s) in each analysis epoch (ES early stable, LS late stable, VL volatile): **d**, **h** 25 WT mice; **e**, **i** 24 *Scn2a*⁺/⁻ mice; **f**, **j** 8 WT mice with chemogenetic inhibition of the corticotectal projection (WT V1-block); and (**g**, **k**), 7 *Scn2a*⁺/⁻ mice with chemogenetic inhibition of the corticotectal projection (*Scn2a*⁺/⁻ V1-block). **, $p < 0.01$ with two-sided Tukey–Kramer test for pairwise comparisons.

environmental uncertainty[4,29]. In particular, using variational Bayes inversion processes, the model infers hidden states that represent beliefs about the sensory input at three hierarchical levels (Fig. 7a; see Methods for details). The lowest level concerns directly on the outcome (aversive versus non-aversive outcomes; $X_1$); the next level on the outcome probability (cue-outcome contingencies; $X_2$); and the

highest level on the likelihood of environmental changes (stability versus volatility; $X_3$). Inference at each level $i$ depends on that at a higher level, and involves the following five parameters: prior predictions on the outcome ($\hat{\mu}_i$); variance of prediction belief ($\hat{\sigma}_i$); learning rates ($\alpha_i$); precision-weighted prediction errors ($\varepsilon_i$); and uncertainties ($unc_i$). In total, the HGF model has 13 parameters that

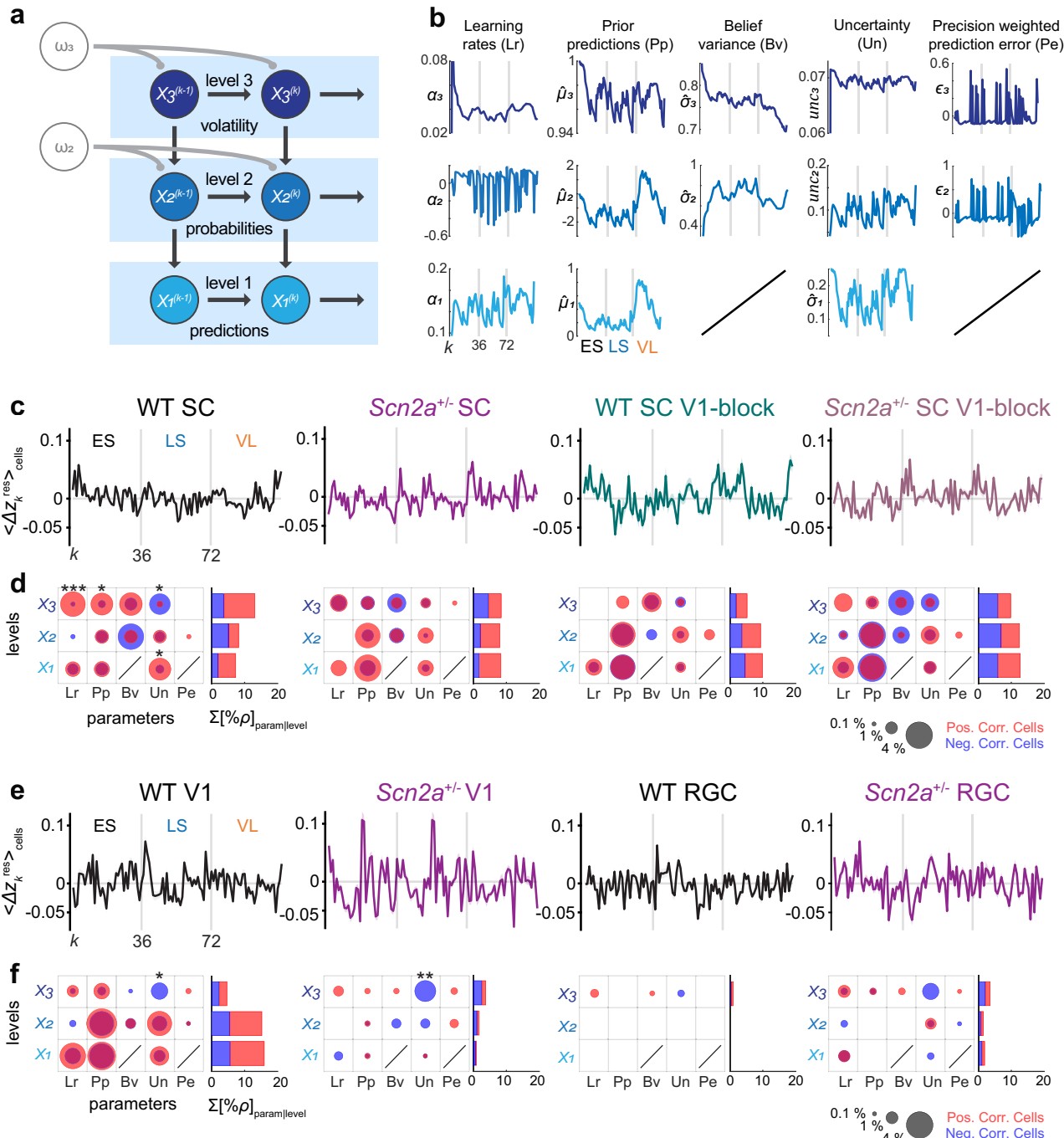

**Fig. 7 | Inputs from the primary visual cortex carry much less information in *Scn2a⁺/⁻* mice than in wild-type mice, leading to misrepresentations of environmental volatility in the superior colliculus.** **a** Schematic diagram of the hierarchical gaussian filter (HGF) model as a Bayesian inference network. Level 1 ($X_1$) represents trial-wise stimulus transitions from one stimulus to the next; level 2 ($X_2$) the probability of the transitions of contingencies; and level 3 ($X_3$) the environment volatility, where $k$ is the current trial number. **b** HGF model parameter trajectories, generated with the cue-outcome contingencies of the implicit visual learning task (Fig. 1; ES early stable, LS late stable, VL volatile). Non-aversive outcome trials were excluded to match neuronal trajectories (in **c**, **e**). Learning rates (Lr) at $X_1$-$X_3$; Prior predictions (Pp) at $X_1$-$X_3$; Believe variances (Bv) at $X_2$ and $X_3$; Outcome uncertainty (Un) at $X_1$; Information uncertainty at $X_2$; Volatility uncertainty at $X_3$; Precision-weighted prediction errors (Pe) at $X_2$ and $X_3$. See Methods for details. **c** Population average of the normalized neuronal trajectories for each experimental group ($\langle \Delta z_k^{res} \rangle_{cells}$; shaded area, s.e.m.). From left to right: wild-type (WT) superior

colliculus (SC), *Scn2a⁺/⁻* SC, WT SC with chemogenetically blocking corticotectal projection (V1-block), and *Scn2a⁺/⁻* SC with V1-block. **d** Neural representation of HGF parameters for each experimental group (as in **c**), given as the percentage of the cells (left) whose response trajectory was significantly correlated with each HGF model parameter trajectory (from **b**), and the cumulative sum of those cell percentages at each level (right; $\sum [\%\rho]_{param|level}$): circle area, percentage of cells with $q < 0.05$ from Storey's false discovery rate (FDR); red, positive correlation; blue, negative correlation. A bias of the correlation polarity for each HDR parameter was examined with a shuffling test with FDR correction: *, $q < 0.05$; **, $q < 0.01$; ***, $q < 0.001$. See also Supplementary Table 2 for details. Corresponding data for the primary visual cortex (V1) and retinal ganglion cell (RGC) axonal responses (**e**, population average and s.e.m. of the normalized neuronal trajectories; **f**, neural representation of HGF parameters). From left to right, WT V1, *Scn2a⁺/⁻* V1, WT RGC, and *Scn2a⁺/⁻* RGC.

are dynamically updated on a trial-to-trial basis (as the two are redundant; Fig.7b; see Methods for details). Each of these parameters provides an insight into distinct aspects of the Bayesian inference process at different levels of uncertainty associated with the sensory input.

To make a fair comparison between the model and neuronal outputs, we focused only on the trials with aversive outcomes because SC neurons were generally not responsive to the non-aversive stimulus in our paradigm (i.e., the omitted stimulus; see, e.g., Supplementary Fig. 3). Thus, after running the simulation with the entire trial sequence, we concatenated the output of the aversive outcome trials alone to generate the trajectories of the HGF model parameters (Fig. 7b). To generate corresponding trajectories of the neuronal signals ($\Delta z_k^{res}$; Fig. 7c, e), we normalized the observed responses of individual neurons for each of the six different types of aversive stimuli, and further filtered out slow non-stationary components (see Methods for details). We then conducted pairwise Pearson's correlation tests with the false discovery rate (FDR) correction between the neuronal and model-parameter trajectories to characterize neuronal representations of the HGF model in the SC of WT and $Scn2a^{+/-}$ mice, respectively. The presence of correlation here does not necessarily mean that these neurons implement the model or directly contribute to Bayesian computation. Nevertheless, it offers an interpretable representation of our experimental data from the hierarchical learning viewpoint. The cue stimulus responses were not considered here because they did not show context-dependent modulation (Supplementary Fig. 2).

We obtained three conclusions from this analysis. First, most of the HGF parameters were represented by some subsets of SC neurons, evidenced by statistically significant correlations between the two trajectories (Fig. 7d and Supplementary Table 2). This was the case with both WT and $Scn2a^{+/-}$ animals, regardless of the chemogenetic inhibition of the corticotectal projection. Nevertheless, SC neurons in WT mice had a distinct representation of the HGF parameters, compared to those in $Scn2a^{+/-}$ animals or when corticotectal projections were chemogenetically blocked. Specifically, WT mice showed a significant polarity bias in the HGF parameter representation, especially at level 3, whereas $Scn2a^{+/-}$ mice showed a tendency towards a positive correlation bias at lower level parameters. Blocking V1 inputs to SC generally attenuated the bias of the parameter representations in both WT and $Scn2a^{+/-}$ animals. This trend was also clear when we computed the cumulative sum of the number of cells correlated with the HGF model parameters at each level. Neuronal trajectories of the SC in WT mice were on average more correlated with level 3 parameter dynamics than those in $Scn2a^{+/-}$ mice or after blocking V1-to-SC projections ($q = 0.03$ and 0.003, respectively; $\chi^2$-test with FDR correction), indicating a misrepresentation of environmental volatility in the SC of $Scn2a^{+/-}$ mice.

Second, V1-to-SC projections in WT mice showed significant correlations with a broad range of HGF parameters across hierarchical levels, while those in $Scn2a^{+/-}$ mice overall showed little correlations (Fig. 7e, f and Supplementary Table 2). In particular, the V1 inputs to SC in WT mice exhibited a significant negative bias for level 3 uncertainties, much as local neurons in SC did. Unlike SC, however, neuronal trajectories of those V1 boutons were more correlated with lower-level parameters ($q < 0.001$, $\chi^2$-test with FDR correction). In contrast, the corticotectal signals in $Scn2a^{+/-}$ mice showed little correlation with any HGF parameter, except for a significant negative bias for level 3 uncertainties. This absence of significant correlations further supports that the V1-to-SC pathway in this ASD mouse model fails to convey contextual information about the visual environment.

Third, hardly any HGF parameter was well represented by the incoming signals from the retina (Fig. 7e, f and Supplementary Table 2). This suggests that retinal inputs have little to do with the contextual learning process, but provide a faithful representation of the visual stimuli.

Taken together, these results provide additional evidence that SC neurons in WT animals process uncertainties about the visual environment by incorporating contextual information from V1. In contrast, SC neurons in $Scn2a^{+/-}$ mice focus primarily on lower levels of uncertainty from recent sensory experiences due to anomalous V1 feedback signaling. This likely causes an oversensitivity to immediate outcomes in ASD[52], highlighting the importance of feedback signaling from V1 in properly processing visual environment context.

## Discussion

Many species have evolved innate behaviors, such as flight or freeze responses, as essential survival strategies to evade predators[53–55]. Mice, for example, respond rapidly and robustly to overhead shadows as they often imply a threat, such as an aerial predator[10,12]. This instinctive response is controlled by the superior colliculus[56–58] (SC), a midbrain structure that directly transforms visual signals from the retina into motor commands, hence bypassing higher cognitive processing in the neocortex[7]. This helps animals react fast to potential threats, prioritizing survival over conscious vision. This can, however, do more harm than good to animals if their response flexibility is too limited, much as in autism spectrum disorder[5,6,50] (ASD). In autumn, for instance, looming shadows arise frequently from falling leaves; however, they are harmless, and thus animals do not need to respond. Adaptive learning allows animals to tailor their behavior according to the specific context, determining properly when to react to or ignore potential threats, such as looming shadows, to ensure survival without unnecessary reactions.

In this study, we dissected the neurophysiological basis of such contextual learning by examining the responses of SC neurons as well as the incoming signals from the retina and the primary visual cortex (V1) in animals engaged in an implicit visual learning task (Fig. 8). In particular, wild-type (WT) mice revealed two key features of the contextual visual integration in the early visual system. The first is a critical role of the top-down signals from V1 to SC. Similar contextual modulation patterns were observed in the aversive stimulus responses of SC cells (Fig. 2) and V1 inputs (Fig. 4), but not retinal ganglion cell (RGC) axons (Fig. 3). Such modulation in SC was eliminated by chemogenetically blocking V1-to-SC projections (Fig. 5). These data support that V1, but not the retina, provides SC with contextual information via the corticotectal projections from layer 5 (L5) pyramidal cells[18,19]. Furthermore, the pupil dynamics of WT mice also demonstrated associated contextual modulation in a way dependent on the V1 inputs to SC (Fig. 6). This suggests that the cortical feedback is central to the adaptive responses at both neurophysiological and behavioral levels.

Another feature of the contextual visual processing is the selectivity of SC to biologically relevant stimuli. While V1 inputs showed adaptation over trials in response to the cue and omitted stimuli, no such change in the response strength was observed in SC or pupil dynamics (Supplementary Figs. 2 and 3). Adaptation in SC cells was observed only to predictable aversive stimuli in a stable environment, likely to prevent an animal's excess reaction; and these SC neurons recovered sensitivity in a volatile environment to regain proper behavioral responses (Figs. 2 and 6). This selectivity indicates that the visual processing in SC is geared to those stimuli that trigger innate defensive behaviors[9–12].

As an ASD model animal to investigate, here we chose SCN2A-haploinsufficient mice for two reasons. First, SCN2A is one of the genes most strongly linked to ASD in humans[24]. Children with mutations in SCN2A have also demonstrated anomalous acoustic responses at the level of electroencephalography[25]. This suggests anomalies in sensory perception, though no direct tests on contextual learning[4] or visual processing have been reported thus far. Second, SCN2A heterozygous knockout is arguably one of the best-established ASD mouse models available[25,26]. It encodes a $Na_V1.2$ subunit that has a clear relevance to neuronal function. Furthermore, $Scn2a^{+/-}$ mice have defects in L5

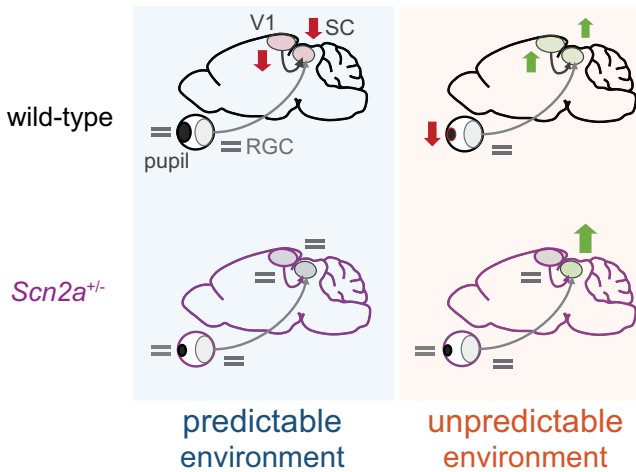

**Fig. 8 | Summary diagram on anomalies in the early visual system of *Scn2a*⁺/⁻ mice during implicit visual learning.** In wild-type mice, top-down inputs from the primary visual cortex (V1) provide critical signals for stimulus-specific context-dependent modulation of the superior colliculus (SC) dynamics during implicit visual learning (down arrow, suppression; up arrow, enhancement). A milder behavioral response, such as associated changes in pupil dynamics, follows that reflects an animal's stimulus pattern recognition. In *Scn2a*⁺/⁻ mice, in contrast, defective V1 inputs lead to anomalous modulation patterns in SC. Bottom-up sensory inputs from retinal ganglion cells (RGC) are largely stable during the task in both wild-type and *Scn2a*⁺/⁻ mice.

corticofugal neurons[26]. These phenotypes made it suitable for addressing how SC cells integrate bottom-up retinal inputs and top-down V1 inputs during contextual visual learning.

As expected from previous studies on ASD subjects[4], *Scn2a*⁺/⁻ animals exhibited anomalous responses during the implicit learning task at both neurophysiological and behavioral levels. In particular, SC neurons demonstrated stable, non-adaptive responses to the aversive stimuli in a stable environment, and an over-enhanced activity in a volatile environment (Fig. 2), while no contextual modulation was observed in the pupil dynamics (Fig. 6). Notably, the V1 inputs exhibited no substantial modulation, either (Fig. 4); and yet, blocking the V1-to-SC projections in *Scn2a*⁺/⁻ mice abolished the atypical modulation pattern of the SC populations (Fig. 5). Taken together with the results from WT animals, we then suggest that the feedback signals from V1 have two counteracting roles in modulating SC responses[18,19]—suppression linked with stability and facilitation with volatility—and their imbalance underlies the anomaly in *Scn2a*⁺/⁻. Specifically, the suppressive function of V1 feedback seems primarily impaired, leading to an imbalance towards facilitation in SC (Fig. 2). It has been shown that reduced expression of Na$_V$1.2 channels in *Scn2a*⁺/⁻ affects signal integration properties of these L5 corticofugal pyramidal cells[26]. They may then fail to properly integrate top-down cortico-cortical signals at the apical dendrites that carry contextual information, especially about environmental stability.

Our probabilistic learning paradigm is closely related to the Bayesian inference principles[1,2]. In particular, one can expect an increase of the prior belief on the stimulus sequence in the stable environment, while a decrease of the likelihood in the volatile environment. Here, we cannot directly relate this to the observed neuronal activity because we do not know exactly how the brain represents those Bayesian probability distributions[51,59]. Nevertheless, *Scn2a*⁺/⁻ mice failed to show adaptive responses in both SC and V1 inputs during the LS epoch (Figs. 2 and 4), while facilitation of SC during the VL epoch remained intact (Fig. 2). This implies that these animals may suffer from a too-weak prior in the learning process, likely taking place in higher-order brain structures, such as the anterior cingulate cortex

whose activity has been suggested to be sensitive to the environmental volatility[60]; and such information is likely transmitted to SC via the top-down modulatory signals.

The above-described scenario was further supported by our hierarchical Gaussian filter (HGF) model analysis (Fig. 7). While the presence of correlations between neuronal and model parameter dynamics does not necessarily mean that these neurons implement the model, we found in both WT and *Scn2a*⁺/⁻ animals that HGF parameters were widely represented across all levels by subpopulations of SC neurons. There was, however, a representation bias at different hierarchical levels: level 3 for WT mice, while level 1 for *Scn2a*⁺/⁻ animals (Supplementary Table 2). This may explain why ASD individuals tend to focus more on immediate sensory inputs and associated outcomes[4,61,62], as they both belong to lower-level HGF inference processes[28,29]. Notably, WT mice showed a transformation of the HGF parameter representations from low-level ones in V1 inputs to high-level ones in SC populations, whereas V1 inputs in *Scn2a*⁺/⁻ animals showed little correlations to the HGF parameters (Fig. 7). If this is indeed the case, resulting sensory overload and/or oversensitivity to unpredictability in ASD[52] may be alleviated by adjusting cortical feedback to bias neural computation towards higher-level HGF inferences in SC, hence redirecting the focus from immediate details to broader contexts. Here, we focused on a specific brain region and a specific computation involved; however, our findings may extend as a more generic computational pattern of ASD within complex, recursive feedforward-feedback interactions of sensory and associative systems. In future studies, it would be interesting to investigate other brain areas in a similar Bayesian framework, and also test different ASD mouse models for better understanding neurophysiological underpinnings of ASD.

## Methods

No statistical method was used to predetermine the sample size. We performed all animal experiments under license 233/2017-PR from the Italian Ministry of Health, following the protocol approved by the Institutional Animal Care and Use Committee at European Molecular Biology Laboratory. We conducted data analyses using Python, Matlab (Mathworks), and R, and reported all summary statistics as mean ± standard error of the mean (s.e.m.) unless specified otherwise. We considered $p$- and $q$-values significant at $\alpha < 0.05$ *, 0.01 **, 0.001 ***, unless stated otherwise.

### Animals

We used C57BL/6J mice (*Mus musculus*, both sex), aged approximately 5–10 weeks at the time of the initial surgery. We housed the mice under a 12-h light/12-h dark cycle and gave them *ad libitum* access to water and food. After the surgical procedures, we grouped the animals based on their operation day. We conducted the imaging experiments when the mice were between 8 and 16 weeks old.

We acquired *SCN2A* haploinsufficient mice as frozen sperm (*Scn2a*) from Riken Bioresource Research Center (RBRC10243), and revitalized the line in a C57BL/6J background. The homozygous mutants (*Scn2a*⁻/⁻) were lethal. We used the heterozygous mice (*Scn2a*⁺/⁻) as a model of autism spectrum disorder[25,26] (ASD), and considered their homozygous littermates (*Scn2a*⁺/⁺) as wild-type (WT).

### Intracranial viral injections

We locally injected pseudotyped adeno-associated viruses (AAVs) into the mouse superior colliculus (SC). These AAVs consisted of AAV2 *rep* and AAV9, AAV5, or AAV2-retro *cap* genes, or a hybrid of AAV1 and AAV2 *cap* genes; and were used to express genetically-encoded calcium indicators (GECIs; jGCaMP7f, jGCaMP8m, or jRGECO1a; Fig. 2a) or Cre-recombinase under the pan-neuronal human synapsin (hSyn) promoter. We also injected pseudotyped AAVs, composed of AAV2 *rep* and AAV8 *cap*, into the primary visual cortex (V1) to express jGCaMP7b

(Fig. 4a) or the inhibitory designer receptor exclusively activated by designer drugs (DREADD), hM4Di-mCherry, under the hSyn promoter[63]. To achieve selective expression of hM4Di in corticotectal V1 neurons, we used an intersectional viral strategy by injecting AAV2retro::hSyn-Cre into SC and Cre-dependent AAV8::hSyn-DIO-hM4Di into V1, thereby restricting DREADD expression to V1 neurons that project to SC via retrograde transport of Cre. The retrograde virus was taken up by axon terminals in SC, transported back to the soma, and expressed in neurons projecting to SC, including those in V1, where the Cre-dependent virus was injected to target the same population. This ensures that only V1-to-SC projecting neurons express the DREADD (Fig. 5a, b).

We performed the intracranial viral injection simultaneously with the cranial implantation, as described below. We created a craniotomy over the right SC or V1 and inserted an injection pipette (with a tip diameter of approximately 30 μm; WPI 1B120F-3 borosilicate glass capillary pulled with Zeitz DMZ puller) filled with a virus solution (ranging from approximately $5 \times 10^{12}$ to $4 \times 10^{14}$ vg mL$^{-1}$ in phosphate-buffered saline, PBS) through the dura. For SC, we positioned the pipette using coordinates relative to Bregma, approximately [−4]–[−4.5] mm anterior-posterior (AP), 0.5–0.7 mm medial-lateral (ML), and slowly advanced it to a depth of about 1.25 mm. For V1, we used the following coordinates: −3.5 mm AP, 1.8–2.0 mm ML, and 0.6 mm dorsal-ventral (DV). We injected the virus solution at a rate of 1–2 nL s$^{-1}$ for SC (0.4–0.6 μL in total) and 0.5–1 nL s$^{-1}$ for V1 (0.25–0.40 μL in total) using a microinjection pump (either Neurostar NanoW or WPI NanoLiter 2010). After completing the injection, we waited at least 10 min before slowly withdrawing the pipette, and then proceeded with the cranial window implantation procedure.

### Intravitreal viral injections

We performed Intravitreal injections of AAV2, pseudotyped with a hybrid of AAV1 and AAV2 capsids, to deliver hSyn-axon-GCaMP6s expression cassette to the mouse retinal ganglion cells[37] (RGCs; Fig. 3a). We exposed the scleral surface on the left eye and made a small piercing with a sterile 28–30 G needle in between the sclera and the cornea. We then inserted an injection pipette (~50 μm tip diameter with 30°–40° bevel) prefilled with a virus solution (~1.5 × 10$^{14}$ vg mL$^{-1}$ in phosphate-buffered saline with 0.001% Pluronic F68 and 0.001% FastGreen) into the vitreous chamber, approximately 1 mm deep. After a good sealing of the pipette was formed, we injected 1.2 μL of the virus solution at a rate of 10 nL s$^{-1}$, using a microinjection pump. We slowly withdrew the pipette at least 5 min after the completion of the injection, and covered the treated eye with the eye ointment. We carried on the rest of the procedure and the head implant as described in the previous paragraph on intracranial injections.

### Cranial implantations

We performed the cranial window implantation over the mouse SC as follows[36,37]. Before the surgery, we prepared a cranial window assembly by activating the surface of a circular glass coverslip (5 mm diameter, 0.13–0.15 mm thickness; Assistent Karl Hecht) with a laboratory corona treater (BD-20ACV Electro-Technic Products). We then fused the activated glass coverslip to a cylindrical silicone plug (1.5 mm diameter, 0.75–1.00 mm height; Kwik-Sil, WPI) by baking it at 70–80 °C for 24 h.

During the implantation, we anesthetized the mice (induction: 4% isoflurane in oxygen; maintenance: 1.5–2.0%) and positioned them in a stereotaxic apparatus (Stoelting 51625). We used a heated plate (Supertech Physiological Temperature Controller) to maintain body temperature and prevent hypothermia, and applied eye ointment (VitA-POS, Ursapharm) to protect the eyes. We disinfected and removed the scalp (Betadine 10%, Meda Pharma), then scratched and cleaned the skull surface to ensure the cement adhered well.

Using a high-speed surgical drill (OmniDrill35, WPI) with a 0.4 mm ball-tip carbide bur (Meisinger), we made a craniotomy of approximately 3.0 mm AP by 2.5 mm ML over the right SC. To prevent bleeding, we applied hemostatic sponges (Cutanplast, Mascia Brunelli) soaked in sterile cortex buffer (NaCl 125 mM, KCl 5 mM, Glucose 10 mM, HEPES 10 mM, CaCl$_2$ 2 mM, MgSO$_4$ 2 mM, pH 7.4) to the craniotomy site. We carefully positioned the cranial window assembly, pushing the transversal sinus and posterior cortex about 0.5 mm forward, with the silicone plug covering the medial-caudal region of the right SC (see, e.g., Fig. 2a). We fixed and sealed the implant with tissue adhesive (Vetbond, 3 M). We then cemented a custom-made titanium head plate (0.8 mm thick) to the skull using a blend of acrylic cement powder (Paladur, Kulzer) and cyanoacrylate adhesive (Loctite 401, Henkel).

After the surgery, the animal was allowed to recover from anesthesia in a warmed chamber and then returned to its home cage. For postoperative care, we administered intraperitoneal injections of Rimadyl (5 mg kg$^{-1}$; Zoetis) and Baytril (5 mg kg$^{-1}$; Bayer) to the mice daily for 3–5 days. We then waited an extra 20–30 days to ensure the cranial window fully recovered before starting in vivo two-photon imaging sessions.

### Habituation

One week before the experiment began, we habituated the mice to both the experimenter and the head-fixation procedure in a dark setting. We positioned the animals on a stable platform (non-rotating disc; about 20 cm diameter) and let them stay still. We mounted a plexiglass tunnel over the rear part of their bodies to increase comfort. We conducted these habituation sessions twice a day, each lasting for 2 h, over five to ten days. This process accustomed the mice to the experimental setup and aimed to reduce any stress during the actual data collection.

### Experimental setup

Two to seven days after training, we placed the animal on a stable platform with its head fixed under a two-photon microscope, in front of a spherical projection screen. We used a non-rotating disc (about 20 cm diameter) as the platform to minimize the effect of motor activities on neuronal firing during the recordings. The platform, however, allowed for vertical excursions, which were compensated by a spring and a counterweight adjustable for each animal and monitored by a Hall sensor. This helped minimize head movements by preventing the mouse from exerting excessive force on the head-holder with its paws. We excluded from the analysis any trials if the average vertical excursion of the platform during the trial exceeded 0.25 mm. We used an infrared (IR) camera (Imaging Source DMK23UX174) with IR illumination to monitor the movement of the animal's left eye during recordings (Fig. 2a; 30 frames per second; 640-by-480 pixels).

### Chemogenetics and histology

We had a subset of mice with hM4Di expressed in V1 (7 WT and 7 Scn2a$^{+/−}$; Fig. 5 and Supplementary Fig. 5). These animals underwent the experiment twice, with at least a two-week interval. For both experiment repetitions, mice received intraperitoneal injections of either a saline solution (0.9% NaCl) containing clozapine N-oxide (0.2 mg mL$^{-1}$; CNO), leading to a final dose of 3 mg kg$^{-1}$ (administered in a volume of 0.1–0.3 mL, adjusted for mouse weight), or the saline alone. We alternated the solution allocation: mice receiving saline alone in the first iteration were given the CNO solution in the second, and vice versa. Starting 45–70 min post-injection, the mice underwent the experiment for approximately 1 h.

Once all the recording sessions were done, we histologically examined the expression of hM4Di-mCherry. We anesthetized the mice terminally with a ketamine/xylazine overdose, and perfused them

intracardially with 4% paraformaldehyde (PFA) in PBS. We then harvested the brains, incubated them for a day in 4% PFA, and for at least another day in PBS. We sliced the fixed brains into coronal sections (100 μm thickness) with a cryostat (Leica CM3050S). We then mounted the slices on a glass coverslip using 4′,6-diamidino-2-phenylindole (DAPI)-mount. We performed fluorescent imaging with a slide scanner (VS200 automated slide scanner, Olympus) with a 10X dry objective (UPLAPO 10x, NA: 0.4, X-line) across three channels: DAPI, GCaMP, and mCherry (Fig. 2b). Only those animals that displayed clear mCherry expression (hence hM4Di expression) in V1 layer 5 (L5) somata (e.g., Fig. 5b) were included in the analysis (Fig. 5c, f and Supplementary Fig. 5a, b; V1-block groups).

## Visual stimulation

We presented visual stimuli to the mice as follows[64]. We used a custom gamma-corrected digital light processing device to project images onto a spherical screen placed approximately 20 cm from the left eye of the animal. The projected images covered the left monocular visual field within ±22° in elevation and from −70° to 0° in azimuth (Fig. 2a). The images were displayed at a resolution of 1280-by-720 pixels and a frame rate of 60 Hz.

One day before the experiment, mice underwent a training process. We first exposed to them a gray screen for at least 10 minutes to adapt to the average luminance in a head-fixed condition under the microscope. Subsequently, we presented the following visual stimuli:

1. Still gratings: A random sequence of black and white square wave gratings in four orientations (0°, 45°, 90°, 135°) with a spatial frequency of 0.1 cycles per degree. Each stimulus lasted for 1 s, followed by a 0.5-s interstimulus interval (ISI). This sequence was repeated 46 times in 5 min.

2. Cue-outcome stable 100%: Horizontal and vertical bars (above-mentioned still gratings with 0° and 90° orientations, respectively) served as cue stimuli. Horizontal bars were followed by an omitted stimulus (prolonged ISI with a gray screen for 2 s), while vertical bars were followed by one out of four possible aversive stimuli (2 s): Sweep-left (a black disk 8° in diameter, appearing at 10° from the top edge and moving horizontally from the left edge to the right one at 42° per second), Sweep-right (moving from right to left), Sweep-left-loom (a Sweep-left for the first second to bring the disk at the upper center of the screen, followed by the disk increasing in size for the next second at a speed of 57° per second while moving its center downward until reaching the horizontal center and a final size of 64° in diameter), and Sweep-right-loom (Sweep-right for the first half, followed by the loom for the second half). To avoid pupillary light reflex, the average luminance of the screen was kept constant at any time. For the sweep and loom stimuli, the background luminance was adjusted for the relative area of the black disk at every frame. We presented three blocks of 72 cue-outcome trials (approximately 5 min), where the stimulus sequence of each block was pseudo-randomized with different seeds.

3. Cue-outcome stable 83%: The cue-outcome probabilities were changed so that horizontal-omitted and vertical-aversive trials occurred with a probability of 83.4%, while horizontal-aversive and vertical-omitted trials occurred with a probability of 16.6%. We presented three blocks of 72 cue-outcome trials (approximately 5 min), where the stimulus sequence of each block was pseudo-randomized with different seeds.

On the day of the experiment, mice were again habituated to the average luminance for at least 10 min. We then presented the following stimuli:

1. Still gratings.
2. Cue-outcome stable 83%, but with six different types of aversive stimuli. Besides the four aversive stimuli described above, we included: Slow-loom (a black 3° disk appeared in the vertical

center at the top edge and increased its size at a speed of 28.5° per second while moving its center downward until reaching the horizontal center and a final size of 64° in diameter) and Fast-loom (a black 3° disk appeared in the vertical center at the top edge and stayed still for 0.5 s; it then increased its size at a speed of 57° per second while moving its center downward until reaching the horizontal center and a final size of 64° in diameter; it then stayed still again for 0.5 s). In total, 216 trials were presented in three blocks (72 trials each; Fig. 1 and Supplementary Fig. 1). Animals typically moved a lot at the beginning and stayed still after a while. Thus, the first block was excluded from the analysis to avoid any movement artifact. The second and third ones were used for the analysis as early-stable (ES) and late-stable (LS) epochs, respectively (Fig. 1b).

3. Cue-outcome volatile: The cue-outcome stimulus sequences were presented with three different contingencies: a given cue-outcome (horizontal-omitted and vertical-aversive) probability of 16.6% for the first 24 trials; 50% for the next 24 trials; and 83.4% for the last 24 trials (Fig. 1c). These data (72 trials in total) were referred to as the volatile (VL) epoch in the analysis. Collectively, the cue-outcome stimuli #2 and #3 constituted an implicit visual learning task.

4. Random water-wave stimuli (3 min): These stimuli were used to generate binary masks for signal source extraction in calcium image analysis (see below Calcium image analysis section).

5. Randomly flickering black-and-white checkerboard stimuli (10 minutes): These stimuli were used for receptive field mapping (Supplementary Fig. 6). The rectangular fields had a width of 3.7° and a height of 2.9°. Each field was independently modulated by white noise at a frequency of 4 Hz.

In the cue-outcome stimuli (#2 and #3), the trials were evenly balanced within each analysis epoch (72 trials each), with 36 horizontal and 36 vertical cues presented before 36 omitted and 36 aversive outcomes. Each of the six different aversive stimuli was presented six times within the 36 aversive trials. The sequences of stimuli in each epoch were pseudorandomized with different seeds, but we used the same set of seeds for all mice. This ensures that each mouse was presented with a new random sequence of stimuli in each epoch, but all mice were presented with the same sets of sequences and cue-outcome contingencies (Fig. 1).

For the subset of mice subject to the chemogenetics experiments (see Chemogenetics and histology section above), we repeated the stimulus paradigm on the same experimental day for CNO or saline treatment.

## In vivo two-photon imaging

During the experiment, we kept the animal with its head fixed inside the rig for a maximum duration of 2 h (typically 2 to 5 imaging sessions per animal). We performed two-photon calcium imaging using a galvo-resonant scanner (Scientifica HyperScope) and SciScan image acquisition software. The microscope setup included a mode-locked Ti:sapphire tunable laser (InSight DS+, Spectra-Physics) and a plan fluorite objective (CFI75 LWD 16X W, Nikon).

In each imaging session, we conducted single-plane time-lapse recordings with a field of view approximately ranging from 0.65-by-0.65 mm to 0.43-by-0.43 mm at a depth of 115–130 μm from the surface of SC. We used the excitation wavelength of 920 nm for GCaMPs, and 1040 nm for jRGECO1a. The average laser power under the objective ranged between 40–80 mW. We bandpass-filtered the emitted fluorescence signal (BP 527/70 or BP 650/100 after beam-splitter FF580-FDi01, Semrock) and detected it using a non-descanned gallium arsenide phosphide photomultiplier tube (Hamamatsu GaAsP PMT). We acquired each frame with either 512-by-512 pixels (16-bit depth) at a rate of 30.9 Hz, or 1024-by-1024 pixels at a rate of 15.5 Hz (e.g., Figs. 2b, 3a, and 4a).

## Calcium image analysis

To simplify manual alignment across different recordings, we first cropped the recorded images from 512-by-512 pixels to 480-by-480 pixels (1.3–0.85 μm per pixel), or from 1024-by-1024 to 960-by-960 (0.65–0.43 μm per pixel) using ImageJ software. We then applied a series of rigid and non-rigid motion corrections in CaImAn[38] (version 8 or 9).

To generate masks for cellular detection in CaImAn, we used either a segment of the recording containing 3000 frames from the random water-wave stimulus presentation period or the entire recording frames. We divided the images into 6-by-6 patches for the constrained non-negative matrix factorization (CNMF) deconvolution, resulting in a total of 36 patches. We set the expected number of neurons per patch (*params.K*) to 5 for SC cell bodies, and set the size of the neurons (*params.gSig*) to 5-by-5 pixels (*half size*) for 512-by-512 acquisitions, or 10-by-10 pixels for 1024-by-1024 acquisitions. For RGC boutons, we set 8 for *params.K* and 2.5-by-2.5 pixels for *params.gSig*. We manually selected putative cells that had a uniformly-filled round shape of approximately 10–20 μm in size, and putative boutons below 2–3 μm. We then converted these selected components into binary spatial masks and performed two iterations of masked CNMF in CaImAn on the entire time-lapse recordings. We used the resulting spatial components (*estimates.A*; e.g., Figs. 2c, f, 3a, d, and 4a, d) and deconvolved neural activities, represented as spike estimations based on the $df/f$ calcium transients (*estimates.S*; e.g., Supplementary Fig. 1), for subsequent analyses.

## Receptive field analysis

The receptive fields (RFs) of SC somata or putative boutons of RGCs or V1 L5 projection neurons were estimated by reverse-correlation methods using the random checkerboard stimuli[37]. Specifically, we calculated the response-weighted average of the stimulus waveform (0.5 s window; 16.7 ms bin width) and characterized its spatial profile by the two-dimensional Gaussian curve fit at the peak latency (e.g., Supplementary Fig. 6a). The RF size was then defined as twice the mean standard deviation of the long and short axes.

Two-way ANOVA (Matlab: anovan) was used to make a comparison across Experimental Group = {WT SC, $Scn2a^{+/-}$ SC, WT SC with V1-block, $Scn2a^{+/-}$ SC V1-block, WT V1, $Scn2a^{+/-}$ V1, WT RGC, $Scn2a^{+/-}$ RGC} and Mouse (41 mice), with the Mouse nested within the Experimental Group. For pairwise comparisons across epochs within each experimental group, we used the Tukey-Kramer tests (R: pairs(emmeans) on aov; Supplementary Fig. 6b)

## Analysis of neuronal activity contextual modulation

First, the estimated spiking activity of each putative neuronal or axonal component in each analysis epoch was extracted from the entire time lapse and converted into z-scores (Matlab: zscore): $Z(t)$. To compare the aversive stimulus response across analysis epochs, we then calculated the average response across trials, $z(t) = \langle Z(t) \rangle_{\text{trials}}$, for each cell for each epoch (e.g., Fig. 2c, d, f, g), excluding those trials with excessive vertical platform excursion (>0.25 mm on average). We then selected responsive cells for subsequent analyses that had $z(t)$ exceeding a predetermined threshold (1 for SC somata; 0.75 for axon terminals of RGCs or V1 cells) in at least one analysis epoch.

For each responsive neuron, we took an average of $z(t)$ over time during the aversive stimulus presentation period (2 s) for each analysis epoch (e.g., Fig. 2d, g): $\bar{z} = \langle z(t) \rangle_t$. We then subtracted the average of $\bar{z}$ across the three epochs for normalization: $\Delta z = \bar{z} - \langle \bar{z} \rangle_{\text{epochs}}$, and used this quantity $\Delta z$ for the population analysis (e.g., Fig. 2e, h). For comparisons across animals, we performed a 3-way analysis of variance (ANOVA; Matlab: anovan) on $\bar{z}$ with a full interaction model. The three variable categories are: Mouse (41 mice), Experimental Group (WT SC, $Scn2a^{+/-}$ SC, WT SC with V1-block, $Scn2a^{+/-}$ SC with V1-block, WT V1, $Scn2a^{+/-}$ V1, WT RGC, $Scn2a^{+/-}$ RGC), and Epochs (ES, LS, VL). The

with the Mouse nested within the Experimental Group. This approach ensures the preservation of the variance structure across groups for assessing the overall effects. For pairwise comparisons across epochs within each experimental group, we used the Tukey-Kramer tests on $\Delta z$ (R: pairs(emmeans) on aov; e.g., Fig. 2e, h). This maintains the nested ANOVA structure for assessing within-group differences. Similar methods were employed to compare the cue or non-aversive stimulus responses across analysis epochs (Supplementary Figs. 2–4).

To characterize baseline response levels of SC cells to the two cue stimuli, for each cell, we first calculated the average responses across vertical or horizontal cue trials during the early-stable epoch, respectively: $z_v(t) = \langle Z(t) \rangle_{t, \text{verticalEStrials}}$ and $z_h(t) = \langle Z(t) \rangle_{t, \text{horizontalEStrials}}$. For those responsive cells to at least one cue stimulus ($\max[z_v(t)] > 1$ or $\max[z_h(t)] > 1$), we averaged these responses over time and computed the difference between them. We then calculated the median of these differences across cells for each animal to quantify the population response bias, and used a Kolmogorov-Smirnov test to compare the distributions of this population response bias between WT and $Scn2a^{+/-}$ mice.

## Pupillometry

Pupil detection and eye-tracking were performed with a region-based convolutional neural network (R-CNN). The network was trained by a set of 260 mouse eye video frames with manually segmented pupil outlines[64]. The detected pupil was parameterized by a 2-dimensional Gaussian curve fit to the pixel map. We then calculated the area of the pupil, $A(t)$, for each frame based on the estimated circular area from the long-axis diameter ($6 \times 10^3$ pixel$^2$ = 1 mm$^2$; e.g., Fig. 6a, b). Note that, assuming a circular pupil on a spherical eyeball, the pupil position affects its short-axis diameter, making a circular pupil ellipsoidal, but the long-axis diameter is invariant. Mann–Whitney $U$-test was used to compare the median pupil area during the implicit visual task between WT and $Scn2a^{+/-}$ mice (Fig. 6c).

To analyze pupil size dynamics across analysis epochs, we first calculated the proportional change in pupil size upon each trial of the aversive stimulus presentation, and then took the average over trials for each epoch: $a(t) = \langle A(t)/A_0 \rangle_{\text{trials}}$, where $A_0$ is the average pupil size over the 10 frames before the stimulus onset for each trial (Fig. 6d–g). We then took the average over the stimulation presentation period (2 s) for each analysis epoch: $\bar{a} = \langle a(t) \rangle_t$, and subtracted the mean across epochs for normalization: $\Delta a = \bar{a} - \langle \bar{a} \rangle_{\text{epochs}}$ (Fig. 6h–k). We performed 2-way ANOVA (Matlab: anovan) to make a comparison across Experimental Group (WT, $Scn2a^{+/-}$, WT with V1-block, $Scn2a^{+/-}$ with V1-block) and Epoch (ES, LS, VL). For pairwise comparisons of $\Delta a$ across epochs at the population level, we used the Tukey-Kramer tests (Matlab: multcompare).

## Hierarchical Gaussian filter model

To simulate how an ideal Bayesian observer would behave in response to the implicit visual learning task, we implemented a hierarchical Gaussian filter (HGF) model[28,29] (Fig. 7a) and calculated its parameter trajectories based on the 216 cue-outcome contingencies (Fig. 7b). The stimuli presented during the experiments followed a pseudo-random sequence that was identical for all animals as described in the Visual stimuli section (Fig. 1). Thus, the sequence of contingencies remained the same across the subjects. For each trial $k$, we assigned a binary value to the contingency: 0 for an expected outcome (vertical-aversive or horizontal-omitted) and 1 for an unexpected outcome (vertical-omitted or horizontal-aversive). These values were represented as a binary vector $u$ that served as the sensory input for the task.

From $u$, the posterior expectations at three levels ($\mu_i$ with $i = 1, 2, 3$) were calculated as follows. At level 1, the posterior is equal to the sensory input: $\mu_1 = u$. At level 2, the posterior expectation about $u = 1$ from the previous trial ($k - 1$) is $\mu_2 = \mu_2^{(k-1)} + \sigma_2^k \delta_1^k$, where $\delta_1^k = \mu_1^k -$

$s(\mu_2^{(k-1)})$ is the prediction error for level 1, with $s(\cdot)$ being a sigmoid function; and $\sigma_2^k = (\hat{\sigma}_1^k + 1/\hat{\sigma}_2^k)^{-1}$ is the standard deviation or variance at level 2. At level 3, the posterior expectation regarding environmental volatility is $\mu_3 = \mu_3^{(k-1)} + \sigma_3^k w_2^k \delta_2^k/2$, where the weight is given as $w_2^k = \exp(\mu_3^{(k-1)} + \omega_2)/(\sigma_2^{(k-1)} + \exp(\mu_3^{(k-1)} + \omega_2))$, the prediction error as $\delta_2^k = (\sigma_2^k + (\mu_2^k - \mu_2^{(k-1)})^2)/(\sigma_2^{(k-1)} + \exp(\mu_3^{(k-1)} + \omega_2)) - 1$, and the variance as $\sigma_3^k = \hat{\sigma}_3^k/(1 + \hat{\sigma}_3^k w_2^k(w_2^k + r_2^k \delta_2^k)/2)$ where $r_2^k = (\exp(\mu_3^{(k-1)} + \omega_2) - \sigma_2^{(k-1)})/(\sigma_2^{(k-1)} + \exp(\mu_3^{(k-1)} + \omega_2))$.

From these posterior expectations $\mu_i$, we derived the prior expectations at each level $\hat{\mu}_i$ that represent the prior prediction about the probability of having $u = 1$ before the sensory input is presented. At level 1, the prior prediction is the sigmoid of the level 2 posterior expectation $\hat{\mu}_1 = s(\mu_2) = 1/(1 + \exp(-\mu_2))$. At level 2, the prior prediction about the probability that $\mu_1 = 1$ is $\hat{\mu}_2 = \mu_2^{(k-1)}$. At level 3, the prior prediction about environment volatility, or probability in change of $\mu_2$, is $\hat{\mu}_3 = \mu_3^{(k-1)}$.

From here, the variance of the belief at every level $\hat{\sigma}_i$, or the inverse of the predicted precision $1/\hat{\pi}_i$, is given as follows. At level 1, the belief variance or imprecision of prediction that $u = 1$ is $\hat{\sigma}_1^k = \hat{\mu}_1(1 - \hat{\mu}_1)$. At level 2, the belief variance about the probability that $\mu_1 = 1$ is $\hat{\sigma}_2^k = \sigma_2^{(k-1)} + \exp(\mu_3^{(k-1)} + \omega_2)$. At level 3, the belief variance about environment volatility, or probability in change of $\mu_2$, is $\hat{\sigma}_3^k = \sigma_3^{(k-1)} + \exp(\omega_3)$.

To calculate the precision-weighted prediction errors $\varepsilon_i$, we first calculated prediction errors at each level: $\delta_i^k = \mu_i^k - \hat{\mu}_i^k$ for $i = 1$; otherwise $\delta_i^k = (\sigma_i^k + (\mu_i^k - \hat{\mu}_i^k)^2)/\hat{\sigma}_i^k - 1$. We then have $\varepsilon_2 = \sigma_2 \delta_1$ and $\varepsilon_3 = \hat{\sigma}_2 \delta_3/\sigma_3$.

The uncertainties $unc_i$ are calculated as outcome uncertainty. This corresponds to the predicted variance $unc_1 = \hat{\sigma}_1^k$ at level 1; the informational uncertainty $unc_2 = s(\mu_2^k)(1 - s(\mu_2^k))\sigma_2^k$ at level 2; and environmental uncertainty $unc_3 = \exp(\mu_3^{(k-1)} + \omega_2)$ at level 3.

Finally, the learning rates $\alpha_i$ are calculated as $\alpha_1 = (s(\mu_2^k) - s(\mu_2^{(k-1)}))/(u - s(\mu_2))$ at level 1; $\alpha_2 = \sigma_2$ at level 2; and $\alpha_3 = \sigma_3 \exp(\mu_3^{(k-1)} + \omega_2)/2\hat{\sigma}_2$ at level 3.

In sum, the HGF model consisted of a set of these 13 parameters in five classes across three hierarchical levels (Fig. 7b): learning rates, $\alpha_1, \alpha_2, \alpha_3$; prior predictions, $\hat{\mu}_1, \hat{\mu}_2, \hat{\mu}_3$; variances of belief, $\hat{\sigma}_2, \hat{\sigma}_3$; uncertainties, $unc_1(= \hat{\sigma}_1), unc_2, unc_3$; and precision-weighted prediction errors, $\varepsilon_2, \varepsilon_3$. After generating the model parameter trajectories from the cue-outcome contingency vector $u$, we eliminated those trials with non-aversive outcomes to make a fair comparison with the neuronal response trajectories that included the responses to aversive outcomes only (see below; Fig. 7c, e).

### Neuronal trajectory estimation

Because we used six different types of aversive stimuli, and because the cue-outcome stimulus patterns were presented in blocks, the observed neuronal responses had a certain systematic bias that needed to be corrected before comparing them to the HGF model parameter trajectories. To this end, we normalized the neuronal responses in two steps: (1) stimulus-specific normalization and (2) non-stationarity elimination. Here, we focused only on the aversive stimulus responses, but not the cue stimulus responses, because the latter did not demonstrate context-dependent modulation in our experimental paradigm (Supplementary Fig. 2).

First, for each neuron, we calculated the average response in z-score for each aversive stimulus trial $\bar{z}_k = \langle Z(t) \rangle_t$, and subtracted the average per stimulus type across three epochs (i.e., 18 trials in total per type): $\Delta z_{k|type} = \bar{z}_{k|type} - \langle \bar{z}_{k|type} \rangle_{trials|type}$. The resulting concatenated response $\Delta z_k$, termed as neuronal trajectory here, represented trial-to-trial differences of the neuronal activity, independent of the stimulus type.

These neuronal trajectories typically showed a slow decay over trials in each analysis epoch, likely due to photobleaching. To eliminate such non-stationary components, we next fitted a power function,

$f(k) = a \cdot k^b + c$ (Matlab: fit power2), to the population average of all the neuronal trajectories in the dataset, averaged across epochs. We then subtracted a scaled decay curve, $d \cdot f(k)$, from the neuronal trajectory of each neuron. The same scalar, $d$, was used across epochs to avoid altering the context-dependent neuronal activity. The residual trajectory ($\Delta z_k^{res} = \Delta z_k - d \cdot f(k)$; Fig. 7c) had a Pearson's correlation with the decay within ±0.05, and was used for the subsequent correlation analyses with the HGF model parameter trajectories.

### Correlation analysis between neuronal trajectories and HGF trajectories

We used Pearson's correlation (Matlab: corr) to examine the relationship between the trajectories of neurons and the HGF model parameters. Specifically, we correlated each of the 13 HGF parameters (Fig. 7b) with every individual neuronal trajectory. The associated $p$-values were then converted to $q$-values with Storey's FDR methods (Matlab: mafdr) to determine the statistical significance. We then counted the fraction of SC neurons significantly correlated with each HGF parameter for each of the four experimental groups (WT SC, *Scn2a*[+/−] SC, WT SC with V1-block, *Scn2a*[+/−] SC with V1-block; Fig. 7d, left panels), which we considered as the neuronal representation of the HGF model. A bias of the correlation polarity was tested with a shuffling test (10,000 repetitions, with FDR correction; Supplementary Table 2). The same procedure was applied to the axonal responses of RGCs and V1 cells (Fig. 7e, f).

At the population level, we also calculated Pearson's correlation, $\rho$, between the population average of the neuronal trajectories $\langle \Delta z_k^{res} \rangle_{cells}$ (Fig. 7c) and each of the HGF parameter trajectories (Fig. 7b). We then took the cumulative sum of the neurons significantly correlated with any HGF parameter at each level to compare how each experimental group represents the HGF parameters across different levels (Fig. 7d, f, right panels).

### Reporting summary

Further information on research design is available in the Nature Portfolio Reporting Summary linked to this article.

## Data availability

The data used in this study are available in the FIGSHARE database under accession code: https://doi.org/10.6084/m9.figshare.28879004 [https://doi.org/10.6084/m9.figshare.28879004]. Source data are provided with this paper.

## Code availability

The code used in this study is available on FIGSHARE along with the relevant data under accession code: https://doi.org/10.6084/m9.figshare.28879004 [https://doi.org/10.6084/m9.figshare.28879004].

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

## Acknowledgements

This work was supported by research grants from EMBL (H.A.) and SFARI (#875447; H.A.); and EMBL Interdisciplinary, International and Intersectorial Postdoctoral Fellowship (EI3POD; L.F.). The EMBL Gene Editing and Embryology Facility is acknowledged for the mouse line rederivation; EMBL Genetic and Viral Engineering Facility for virus production; EMBL IT Support for provision of computer and data storage servers; and the LAR facility for taking care of animals. We thank all the Asari lab members as well as Cornelius Gross, Eleonora Satta, and Michael Lombardo for many useful discussions.

## Author contributions

L.F. and H.A. designed the study; L.F. performed experiments and analyzed the results; L.F. and H.A. wrote the manuscript.

## Funding

## Competing interests

The authors declare no competing interests.
