## [Transparent Peer Review file · Nature Communications]

Atypical cortical feedback underlies failure to process contextual information in the superior colliculus of *Scn2a*^{+/-} autism model mice

Corresponding Author: Dr Hiroki Asari

Version 0:

Reviewer comments:

Reviewer #1

(Remarks to the Author)

In this manuscript, Ferrarese and Asari propose that SCN2a heterozygotes mice present an atypical cortical feedback that underlies failure to process contextual information in the Superior Colliculus (SC). This is an interesting topic and the author made good efforts to interpret their results using a Bayesian Inference framework. While the task and experiments are well designed, I believe that a major experiment would greatly improve the manuscript.

Major comments:

- 1- The authors talk about cortical feedback to SC and its role in contextual processing in SC. Despite the perturbation of visual cortex (VC) projections to SC using chemogenetics, these experiments would be strengthened by imaging VC-projecting boutons directly to determine if it is the signal coming from VC, the processing of VC inputs in SC neurons, and/or both that are affected. This experiment must be done in both WT and SCN2A het mice.
- 2- The author reported a mechanism involving V1 projecting neurons to SC. However, the coverage of their chemogenetic manipulations is unclear. For example, on Figure 4, the expression of hM4Di does not cover V1 and is very medial and even potentially higher visual areas such as AM. To understand the effect of L5 projecting neurons to SC in this task, the author should quantify the expression of their viral expression and evaluate if, depending on the trial (in/out the receptive field visual stimulus), impacts differently the activity in SC and the behavior. Also, why are the authors not presenting the heat maps per condition in Figure 4 (such as they did in 3B, 2D, 2G, for example)?
- 3- Figure 2 would highly benefit from quantification of the z-score at the single cell level. How many neurons are modulated? Which part of the visual response is affected comparing WT and SCN2A hets: Is it more the onset/offset response or the plateau response? My point is that, by eye, it is hard to grasp if this difference in averaged response is due to a small fraction of neurons or most responding neurons (suppressed or excited).
- 4- Can the author discuss the absence of onset response and the fact that only half of the boutons responded to visual stimulation during the task in Figure 3? Does it mean that the onset response is only inherited from VC? Related to this point and point 3, by looking at Supp Fig 2, by eye, it is mainly onset that is affected. The author should provide more quantification and discuss these results that might be relevant for this study.
- 5- To interpret a difference in pupil diameter, the authors have to make sure that it is quantified into the same conditions, ie, for the same position of the center of the pupil in the skull. This is critical as the eyeball is circular and depending on the position of the pupil, the pupil diameter will vary. So first, the author should quantify the frequency of eye movements comparing the 4 conditions in Fig 5B. Second, compare the average median pupil diameter for similar position of the eyeball. Note that I assume that here, the position of the camera compared to the eyeball is similar between animals.

More minor comments:

- L2, the author should immediately talk about ASD and not autism
- L97, retinal and not regional.
- Are the authors using the same ranking for their heatmap in general. How do they rank their neurons from one condition to another (all figures)
- Is the basal activity without stimulation affected in WT, SCN2a het and V1 perturbation experiments?
- Fig Sup 1 should also compare ES response in WT and SCN2a hets.

- L247 Projecting and not projected
- Can the author discuss that their control chemogenetic experiments are performed using Saline and not CNO in GFP injected mice for example. As CNO has been reported to affect "brain state" related to attention, it would help if the author can provide references showing that CNO has no effect at this concentration.

Reviewer #2

(Remarks to the Author)

This paper aims to investigate how neural activity in the Superior Colliculus (SC) in mice contributes to Bayesian belief updating. The authors present a novel finding that autism model mice exhibit atypical feedback signals from V1 to SC, leading to impaired associative learning in a dynamic environment. The experimental design is comprehensive, involving wild-type (WT) and autism model mice, virus injection, and chemogenetic manipulation to dissect neural circuits. The inclusion of monitoring pupil dynamics is also commendable. However, I have some concerns regarding the interpretation of the results and the conceptual framework used in the study. Here are my suggestions for improving the paper:

1. Conceptual framework for the Bayesian model: While the authors have built a hierarchical Bayesian model to test with neural data, it is important to note that their framework reflects only one possibility of implementing Bayesian integration in the brain. Specifically, implementing prior as top-down feedback and likelihood as bottom-up input is only one possibility of realizing Bayesian integration in the brain (Lee & Mumford, 2003). There are other approaches, such as spiking activities in a population code (Ma et al., Nat. Neuro., 2008), sampling-based representations (Berkes, Orban, Fiser, Lengyel), and recurrent computations (Sohn et al., Neuron., 2019). Including a discussion of these alternative models and their relevance to the findings would enhance the paper's contribution and broaden its readership (Walker et al., Nat. Neuro., 2023; Sohn & Narain, Curr. Opin. Neuro., 2021).

2. Model prediction: The visualization of the model predictions in Figure 1D and 1E is confusing. It would be helpful if the authors could present the model predictions in a way that directly relates to the neural activity differences between early stages (ES), late stages (LS), and very late stages (VL) epochs, as shown in later figures. Currently, it is not clear what critical differences the two autism models make in the LS and VL epochs. Additionally, the text describing the model prediction is also unclear, particularly regarding the dominance of the likelihood in model 1 (line 144; overly high likelihood will dominate posterior and posterior will follow likelihood, not prior). Clarifying this point would improve the paper's clarity.

3. Interpreting SC activities in terms of Bayesian models: The paper's interpretation of SC neural activity in relation to the hierarchical Bayesian model is confusing. The authors state that SC encodes learning rate and uncertainty, but this does not align with the activity difference pattern observed between ES, LS, and VL epochs. In the early results section, the authors interpret the low SC activity during LS as indicating lower prediction error, which contradicts the hierarchical Bayesian model comparison. It is important to clearly distinguish between bias (posterior mean - prior/likelihood) and variance (or precision) when linking them to neural data, as they play distinct computational roles in Bayesian models.

Another question that arises is why SC is considered suitable for computing posterior distributions in the Bayesian sense. The fact that most SC neurons did not respond to omitted aversive stimuli is not consistent with the idea that SC is responsible for posterior computation. Since the horizontal cue still has a probability (17%) of predicting aversive stimuli, SC should generate a response if it were responsible for posterior computation. This discrepancy needs to be addressed and explained.

Furthermore, it is crucial to note that the Bayesian model is applied to the conditional probability $p(\text{outcome}|\text{cue})$, not $p(\text{cue})$ alone. Therefore, the Bayesian inference should be carried out in high-level sensorimotor regions, rather than in the sensory domain. This has implications for the conceptualization and terminology used in the paper. For example, the posterior output does not lead to a "perception" error, as depicted in Figure 1.

4. Neural data analysis: The majority of the analysis in the paper focuses on the period when aversive stimuli are presented. However, both at the Bayesian model and behavioral levels, the cue period also deserves thorough analysis. During the cue period, the prior prediction is likely formed and maintained until the outcome period, where it is integrated with actual outcome information. Analyzing the cue period and its relation to the outcome period would provide valuable insights (Darlington et al., Nat. Neuro., 2019 and Walker et al., Nat. Neuro., 2020). It would also be beneficial to provide a better characterization of visual responses in SC, as any differences in simple visual responses between WT and Scan2a mice could be critical. Additionally, exploring the effects of different types of aversive stimuli (looming versus sweeping, direction of sweeping) would further enrich the findings.

5. Minor points:

Distinction of early versus late stable epochs is rather arbitrary. It needs to be justified (as even the 1st block was removed due to movement artifacts). More fine-grained analysis of VL epoch can also reveal something interesting.

The proportion of neurons correlated with the Bayesian model variables seem too low even though they are statistically significant. Given the total number of neurons is a few hundreds, it must be only a few neurons.

Pupil data was not fully explored in that it was not directly analyzed together with neural data. For example, how is it related to Bayesian model variables? Can it be linked to neural data in a trial-by-trial manner?

Reviewer #3

(Remarks to the Author)

Farrarese and Asari performed calcium imaging, pupillometry, chemogenetics and a hierarchical Bayesian learning model to investigate atypical contextual visual integration in *Scn2a*^{+/-} mice engaged in an implicit learning task in stable and volatile environments. The results suggest that the superior colliculus (SC), not retinal inputs to SC, drives the observed distinct context-dependent modulation of neuronal and pupillary responses of wild-type and *Scn2a*^{+/-} mice. Significantly, the authors propose that V1 L5 pyramidal neurons likely mediate contextual visual integration and this corticotectal pathway may underlie abnormal sensory contextual learning in autism spectrum disorder (ASD). There are significant gaps, unfortunately, that make it difficult to evaluate the data fully, largely because data are piecemeal between WT and *Scn2a*^{+/-} mice, with some major aspects only done in WT mice. A more balanced comparison between the genotypes would strengthen the work considerably.

1) Implicit in the experimental design (and the framing of the work) is the idea that NaV1.2 has a major role in the corticofugal neurons that feed into SC. But the effects of loss of NaV1.2 in other components of the SC integrative pathway are not considered. NaV1.2 appears to be expressed in retinal ganglion cells (PMID 31734278). Is it expressed in SC as well? These components need to be considered before making conclusions that the effects observed are purely mediated by some “top down” effect from V1.

2) An understanding of how the pupillometry measurements (and the calcium imaging data) relate to the stimuli is very opaque. Furthermore, consistent experiments across genotypes would be very useful for determining where (and when) deficits arise in *Scn2a*^{+/-} conditions. The following need to be addressed:

2a) a thorough analysis of these measurements relative to sweeping and looming spots is critical to understand how the genotypes behave in response to stimuli. Do the *Scn2a* heterozygous mice respond consistently on each trial? Are they as attentive over time? Does the response adapt over time?

2b) Some evaluation of visual acuity in the *Scn2a* mice is required, especially because many children with *SCN2A* loss are diagnosed with cortico-visual impairment. At the very least, experiments described in Fig 3 should also be done in *Scn2a*^{+/-} mice.

2c) Pupil time series traces should be presented with the accompanying calcium time series traces in each figure. Instead of (or perhaps in addition to) plotting $\langle z(t) \rangle$ for the aversive stimuli of each epoch, a plot of the average neuronal response pre/post presentation of the neutral cue and pre/post aversive cue onset in the same time series should be included. I ask because these data may reveal the anticipatory response of wild-type and *Scn2a*^{+/-} to the forthcoming non-aversive or aversive stimulus following the neutral cue and glean insight into the overall engagement of the mice with the task.

3) Was there a baseline level activity difference in SC neurons when exposed to the neutral, non-aversive, vertical and aversive stimuli between wild-type and *Scn2a*^{+/-} mice? In looking at the data from WT mice in Supplementary Fig. 1, SC neuron calcium activity skewed more active when presented with vertical gratings preceding the aversive stimuli. Was this comparable in the *Scn2a*^{+/-}? Are the receptive fields different? If so, one might expect a bias in task performance based on perception?

4) Taking a step back, why was *SCN2A* chosen amongst all ASD-related genes? ASD is a spectrum, and it is unclear whether children with *SCN2A* loss of function present with issues as described in Lawson 2017. Have such studies been done within this particular group?

5) The control for DREADDs is incorrect. One needs a sham virus with CNO, not saline, given the well-known off target effects of CNO across the brain.

6) It is unclear how statistics were done with large populations of cells within a relatively smaller population of mice. There will be inter-animal variability; was this taken into consideration? I.e., were the data analyzed in a nested fashion? It was unclear whether this was the case from the text provided.

Minor:

Fig 4b does not show that injections in V1 reach SC. A supplemental image showing mCherry in SC is important to show. Furthermore, it is critical to know how large of a population of V1 cells is labeled across genotypes to interpret properly the hM4Di results.

Methods: “Analysis of neuronal activity contextual modulation” - How was the threshold value determined for selecting responsive cells for subsequent analyses? Based on this thresholding, a range of 22 to 99 cells does not seem balanced, or I may misunderstand.

Line 126 “we presented aversive outcome stimuli more frequently after one neutral cue stimulus than after another.” Rephrase for clarity, please.

Version 1:

Reviewer comments:

Reviewer #1

(Remarks to the Author)

The authors have addressed my previous concerns with additional experiments and analysis, providing convincing responses. Most of my initial concerns stemmed from misunderstanding aspects of the experimental protocol and analysis. However, I remain unclear on two key points that would benefit from clarification in the methods section:

Silencing coverage: Why was optogenetic silencing applied to such a small portion of the visual field (median 22% of V1)? Does the author look exactly at the same cells in SC? This limited coverage may lead the authors to underestimate their results.

Receptive field analysis: Regarding my previous major comment 3 about potentially underestimating the number of modulated neurons - if the authors mapped neuronal receptive fields, shouldn't the analysis only include neuronal responses when visual stimulation falls within each neuron's receptive field? This methodological detail needs clarification.

Given that several of my original concerns arose from protocol ambiguities, I recommend the authors provide more detailed methodological descriptions to prevent similar misunderstandings for future readers.

Reviewer #2

(Remarks to the Author)

The authors have made a commendable effort to address the reviewers' feedback, with new experiments enhancing the specificity of the V1-to-SC projection's changes in different task environments.

However, I have reservations regarding the use of the Bayesian framework in the paper. While the approach itself is not incorrect, the Bayesian model may detract from the paper's strengths, which lie in the detailed circuit dissection and well-designed behavioral tasks. Specifically, I recommend removing Figure 1c and related text from the Results section for several reasons. Firstly, Figure 1c is only conceptual and does not directly generate predictions on the neural data ('high-low-high' activity across 'ES-LS-VL' epochs), leading to ambiguity in its interpretation. Questions arise, such as the representation of likelihood and prior - are they related to aversiveness, perception error, or cue probability? The hierarchical Bayesian model in later figures provides a more detailed and precise understanding of multiple neural signals. Therefore, removing Figure 1c would not diminish the paper's quality.

Secondly, Figure 1c and its text do not address bias or prediction error, which are crucial first-order statistics when considering variance, a second-order statistic. The Bayesian model's core idea involves a bias-variance trade-off, and focusing solely on variance is incomplete.

Thirdly, the text accompanying Figure 1c appears to offer post-hoc explanations to fit the data, rather than presenting a quantitative model. Questions arise, such as why the volatile environment "lowers likelihood" or why likelihood and prior are assumed to be Gaussian distributions. Did animals have more sensory noise for aversive or non-aversive stimulus? And why does "the prior belief remained relatively high and precise because updating the prior is slow in general"? Didn't animals learn that cue-outcome contingency from late stable epoch is not valid anymore in a volatile environment?

An alternative to Figure 1c could be to generate predictions on 'outcome uncertainty'. In the early stable environment, the outcome distribution is bimodal, with equal aversive and non-aversive outcomes. Through learning in the stable epoch, this distribution becomes skewed, leading to reduced outcome uncertainty (similar to posterior variance). In the volatile environment, WT animals would learn a less skewed bimodal distribution, resembling the early stable epoch. This approach does not require the Bayesian framework and offers a more parsimonious explanation. The authors could still fully utilize the Bayesian model later with Figure 7.

Ultimately, the decision on Figure 1c rests with the authors. Congratulations on the work.

Reviewer #3

(Remarks to the Author)

The authors have performed additional experiments and analysis that improve the manuscript considerably. There is one aspect that should be amended. In response to concern #3: Was there a baseline level activity difference in SC neurons when exposed to the neutral, non-aversive, vertical and aversive stimuli between wild-type and Scn2a^{+/-} mice? The authors should make direct comparisons demonstrating that responses to horizontal and vertical cues differ between WT and Scn2a^{+/-} rather than relying on population averages. These are critical values that are needed to better interpret the calcium responses to aversive stimuli.

Version 2:

Reviewer comments:

Reviewer #1

(Remarks to the Author)

Thank you for your responses.

Regarding my two previous comments:

1- I would suggest clearly stating the coverage and providing a thorough discussion of this aspect.

2- If mapping was performed, you should be able to determine when the stimulus enters the receptive field (RF). The surround RF activation is likely minimal using this protocol, and spontaneous eye movements in well-habituated/trained subjects are also minimal. Numerous studies conducting mapping have demonstrated that it does not significantly affect the response for a given firing rate (on average). If you determine that you cannot perform this analysis, you should at minimum discuss this limitation and/or clearly state your experimental conditions.

Reviewer #2

(Remarks to the Author)

The modifications the authors made address my comments.

Reviewer #3

(Remarks to the Author)

Thank you for the clarification and for providing updated information regarding baseline responses in SC. I have no further concerns, and congratulate all the authors on an impressive work.

We thank the reviewers for their constructive comments (in blue). Below are our point-by-point replies (in black).

Reviewer #1 (Remarks to the Author):

In this manuscript, Ferrarese and Asari propose that SCN2a heterozygotes mice present an atypical cortical feedback that underlies failure to process contextual information in the Superior Colliculus (SC). This is an interesting topic and the author made good efforts to interpret their results using a Bayesian Inference framework. While the task and experiments are well designed, I believe that a major experiment would greatly improve the manuscript.

Major comments:

1- The authors talk about cortical feedback to SC and its role in contextual processing in SC. Despite the perturbation of visual cortex (VC) projections to SC using chemogenetics, these experiments would be strengthened by imaging VC-projecting boutons directly to determine if it is the signal coming from VC, the processing of VC inputs in SC neurons, and/or both that are affected. This experiment must be done in both WT and SCN2A het mice.

Following the reviewer's suggestion, we recorded the activity of SC-projecting V1 neurons at their synaptic terminals in SC, in both WT and Scn2a^{+/-} mice engaging in our implicit visual learning task (**Figure 4** in the revised manuscript). In WT mice (n=8), the input from V1 and local SC neurons showed similar patterns of contextual modulations: i.e., adaptation at the Late Stable (LS) epoch and the recovery at the Volatile (VL) epoch. Our HGF model analysis, however, revealed that V1 input largely conveyed lower-level information (levels 1 and 2), whereas SC cells mainly carried higher-level information (level 3; **Figure 7** in the revised manuscript). In Scn2a^{+/-} mice (n=7), in contrast, V1 input to SC showed no contextual modulation (while SC neurons showed over-sensitization in VL) and contained much less information than SC cells. Together with the results of the chemogenetic perturbation experiments (**Figure 5** in the revised manuscript) and the recordings of the retinal inputs (**Figure 3** in the revised manuscript), these results indicate that cortical signals in Scn2a^{+/-} mice fail to transmit contextual information to SC. Note that 1) this may arise from defects in V1 and/or higher cortical areas; and 2) it remains unclear if the processing of SC itself is altered or not. It is a future challenge to address these questions.

2- The author reported a mechanism involving V1 projecting neurons to SC. However, the coverage of their chemogenetic manipulations is unclear. For example, on Figure 4, the expression of hM4Di does not cover V1 and is very medial and even potentially higher visual areas such as AM. To understand the effect of L5 projecting neurons to SC in this task, the author should quantify the expression of their viral expression and evaluate if, depending on the trial (in/out the receptive field visual stimulus), impacts differently the activity in SC and the behavior.

We quantified the viral expression patterns from histology data, and confirmed that the targeted projection originated mostly from the medial-anterior portion of V1 layer 5. Labelled cells occupied 31±22% (mean ± standard deviation; median, 22%) of V1 L5 area, and projections were clearly visible in the ipsilateral SC in all 7 WT and 7 Scn2a^{+/-} mice examined (e.g., **Figure 5b** in the revised manuscript).

As the reviewer pointed out, the chemogenetic viral expression was also detected in V1 layer 2/3 and V2ML/MM areas (e.g., **Figure 5b** in the revised manuscript). We think that this non-specific expression arose from a leak of virus from the injection needle during insertion/retraction procedures. For instance, V1 layer 2/3 cells were also labelled with DIO-hM4Di-mCherry, likely because they project to V2MM area where we had a leakage of AAV2retro-Cre virus (as indicated by co-injected AAV9-GCaMP signals). Nevertheless, 1) these cortical neurons do not project to SC to the best of our knowledge; and 2) they may interact with SC-projecting neurons in V1 layer 5, but here we aimed to inhibit these corticotectal neurons anyway. Therefore, inhibition of upstream cortical neurons should not affect our conclusion on the role of V1-to-SC projection.

Also, why are the authors not presenting the heat maps per condition in Figure 4 (such as they did in 3B, 2D, 2G, for example)?

The corresponding activity heatmap for Fig. 4 was included in the Supplementary Fig. 2B,C in the original manuscript. To avoid confusion, we moved it to the main figure (**Figure 5c,e**) in the revised manuscript.

3- Figure 2 would highly benefit from quantification of the z-score at the single cell level. How many neurons are modulated?

Here we found that ~6% of WT SC neurons (N=67 out of 1080 cells) showed a significant modulation across epochs ($p < 0.05$ after FDR correction; **Supplementary Table 1** in the revised manuscript). Likewise, relatively small fractions of the recorded units were significantly modulated in WT V1 axons (~7%), *Scn2a*^{+/-} SC cells (~4%), and *Scn2a*^{+/-} V1 axons (~5%). These are, however, underestimated values because not all stimuli fell in the receptive field of a given cell across all trials; hence, trial-to-trial variability was inevitably high in our datasets. Many cells indeed showed a tendency of modulation (e.g., **Figure R1**), and the activity (z-score) distribution was wide and unimodal (**Supplementary Figure 3B** in the revised manuscript). This is why we focused on the analysis at the population level, instead of the single-cell level, to reveal the modulation patterns in our experimental paradigm.

Figure R1: Distribution of the activity difference in the responses of WT SC cells between analysis epochs (from left to right: LS vs. ES, VL vs. LS, VL vs. ES). We observed a weak modulation across neurons, instead of a strong modulation in a small fraction of the cells.

Which part of the visual response is affected comparing WT and SCN2A hets: Is it more the onset/offset response or the plateau response? My point is that, by eye, it is hard to grasp if this difference in averaged response is due to a small fraction of neurons or most responding neurons (suppressed or excited).

Because of the retinotopy, 1) each aversive stimulus drove cells at different timings in our experimental design; and 2) each cell showed distinct response patterns to different aversive stimuli (e.g., **Figure R2**). It is thus difficult to analyze the temporal dynamics of the response modulation at the single-cell level, or clearly distinguish between onset and offset responses in the population analysis. To address the reviewer's point, we nevertheless divided the analysis window into the first and second half in time (**Supplementary Figure 4** in the revised manuscript; further dividing the analysis window was not helpful due to high trial-to-trial variability). We found that the modulation was relatively stronger for the later components of the population responses of the SC cells and V1 inputs in WT mice. In contrast, the contextual modulation was rather stronger for the earlier response components in *Scn2a*^{+/-} mice, or after chemo-genetically blocking V1 inputs to SC in WT mice.

Figure R2: Example dynamics of SC somata (a), RGC boutons (b), and V1 axonal boutons (c) in response to different aversive stimuli used in this study (top, merged; bottom panels, different stimuli). Cells were sorted for their positions in the SC image plane along the ML axis, clearly demonstrating that these cells responded at different timings because the stimulus hit their receptive fields at different timings, following the retinotopy.

4- Can the author discuss the absence of onset response and the fact that only half of the boutons responded to visual stimulation during the task in Figure 3? Does it mean that the onset response is only inherited from VC? Related to this point and point 3, by looking at Supp Fig 2, by eye, it is mainly onset that is affected. The author should provide more quantification and discuss these results that might be relevant for this study.

Multiple factors, including both biological effects and experimental artefacts, would underlie the difference in the measured response properties between SC somata, RGC boutons, and V1 axonal boutons.

The first is the difference in the imaging field of view. To gain high enough spatial resolutions, we had to choose a smaller field of view for axonal bouton imaging than for somatic imaging (by a factor of 1.5). Because the sweeping stimuli started and ended at the screen border, the onset/offset of these stimuli did not fall in the receptive field of all the RGC boutons, hence not triggering their responses. Moreover, the recorded cells were evoked at different timings by these dynamic stimuli due to retinotopy (e.g., **Figure R2**). Thus, we did not distinguish between onset and offset responses in our analysis, though we found certain bias in the strength of contextual modulation over time (as mentioned above; **Supplementary Figure 4** in the revised manuscript).

Another reason for the difference in the response strength across brain areas resides in the visual response properties of the cells. For example, RGCs in general have smaller receptive fields than SC somata or V1 cells (**Supplementary Figure 6** in the revised manuscript). This led to a lower chance for our visual stimulus to fall into their receptive fields, even though we adjusted the position of the stimulation screen to cover the receptive field of as many recorded cells as possible. Moreover, while RGCs responded equally well to all aversive stimulus types (e.g., **Figure R2b**), SC somata responded more strongly to the looming stimuli (e.g., **Figure R2a**) and V1 axons to sweeping stimuli (e.g., **Figure R2c**). Such differences in the cells' tuning properties may cause a certain bias in the population response dynamics during the stimulus presentation period. However, we do not have enough trial numbers to assess the contextual modulation on each aversive stimulus due to high trial-to-trial variability.

5- To interpret a difference in pupil diameter, the authors have to make sure that it is quantified into the same conditions, ie, for the same position of the center of the pupil in the skull. This is critical as the eyeball is circular and depending on the position of the pupil, the pupil diameter will vary. So first, the author should quantify the frequency of eye movements comparing the 4 conditions in Fig 5B. Second, compare the average median pupil diameter for similar position of the eyeball. Note that I assume that here, the position of the camera compared to the eyeball is similar between animals.

We fully acknowledge the reviewer's point. Assuming a circular pupil on a spherical eyeball, the pupil position indeed affects its short-axis diameter, making a circular pupil ellipsoidal in the recorded images. However, the long-axis diameter is invariant. We thus re-analyzed the pupil size modulation based on the estimated circular area from the long-axis diameter, and reached the same conclusion as before (based on the ellipsoidal area in the original manuscript). The position of the camera was roughly the same across the recordings. We updated the figure accordingly (**Figure 7** in the revised manuscript). We also included the corresponding pupil dynamics whenever we showed neuronal dynamics in the revised figures.

More minor comments:

- L2, the author should immediately talk about ASD and not autism

We followed the suggestion and revised the manuscript accordingly.

- L97, retinal and not reginal.

We fixed the typo in the revised manuscript.

- Are the authors using the same ranking for their heatmap in general. How do they rank their neurons from one condition to another (all figures)

Yes. To generate an activity heatmap, we sorted the cells based on the response strength in the ES epoch, and used the same order for the other conditions (LS and VL epochs).

- Is the basal activity without stimulation affected in WT, SCN2a het and V1 perturbation experiments?

No. We considered the neuronal activity during the omitted stimulus period as the baseline, and found no notable change in the SC cells' activity in both WT and Scn2a^{+/-} mice. In general, these cells were non-responsive to the omitted stimulus (e.g., **Supplementary Figure 1** in the original/revised manuscript), and stayed silent without visual stimuli regardless of V1 inputs (**Supplementary Figure 3a,b** in the revised manuscript). Unlike SC cells, V1 and RGC axons demonstrated a certain level of responses to the omitted stimulus (**Supplementary Figure 3d** in the revised manuscript). However, no contextual modulation was found across the analysis epochs, except for WT V1 axonal responses (**Supplementary Figure 3e** in the revised manuscript). The pupil dynamics did not show significant contextual modulation, either, during the omitted stimulus presentation, regardless of the chemogenetic V1 perturbation (**Supplementary Figure 3c** in the revised manuscript).

- Fig Sup 1 should also compare ES response in WT and SCN2a hets.

We included the ES response examples from Scn2a^{+/-} mice as well in **Supplementary Figure 1** of the revised manuscript. Please note that this figure only illustrates typical response examples from the two mouse lines for a reference purpose. The outcome of proper quantifications is shown in the following figures in the revised manuscript.

- L247 Projecting and not projected

We updated the figure title in the revised manuscript.

- Can the author discuss that their control chemogenetic experiments are performed using Saline and not CNO in GFP injected mice for example. As CNO has been reported to affect "brain state" related to attention, it would help if the author can provide references showing that CNO has no effect at this concentration.

Thanks for pointing out the caveat on the effect of CNO itself. We performed proper control experiments in WT mice: i.e., 7 animals expressing hM4Di receptors in V1, with versus without

CNO injection (**Figure 5c,d** versus **Supplementary Figure 5a,b** respectively); and 8 mice without hM4D expression but with CNO injection (**Supplementary Figure 5c,d**), of which N=6 animals were tested without CNO injection as well (**Supplementary Figure 5e,f**). We confirmed that CNO injection or hM4D expression itself had no effect on the contextual-modulation in SC cells of WT mice (see also **Supplementary Table 1**).

Reviewer #2 (Remarks to the Author):

This paper aims to investigate how neural activity in the Superior Colliculus (SC) in mice contributes to Bayesian belief updating. The authors present a novel finding that autism model mice exhibit atypical feedback signals from V1 to SC, leading to impaired associative learning in a dynamic environment. The experimental design is comprehensive, involving wild-type (WT) and autism model mice, virus injection, and chemogenetic manipulation to dissect neural circuits. The inclusion of monitoring pupil dynamics is also commendable. However, I have some concerns regarding the interpretation of the results and the conceptual framework used in the study. Here are my suggestions for improving the paper:

1. Conceptual framework for the Bayesian model: While the authors have built a hierarchical Bayesian model to test with neural data, it is important to note that their framework reflects only one possibility of implementing Bayesian integration in the brain. Specifically, implementing prior as top-down feedback and likelihood as bottom-up input is only one possibility of realizing Bayesian integration in the brain (Lee & Mumford, 2003). There are other approaches, such as spiking activities in a population code (Ma et al., Nat. Neuro., 2008), sampling-based representations (Berkes, Orban, Fiser, Lengyel), and recurrent computations (Sohn et al., Neuron., 2019). Including a discussion of these alternative models and their relevance to the findings would enhance the paper's contribution and broaden its readership (Walker et al., Nat. Neuro., 2023; Sohn & Narain, Curr. Opin. Neuro., 2021).

We never meant that SC directly contributes to Bayesian inference, and sincerely apologize for the confusion. Correlations between neuronal dynamics and model parameter dynamics do not necessarily mean that these neurons implement the model.

Our goal here is to better understand ASD-associated neurophysiological phenotypes by characterizing sensory anomalies of ASD model mice in a similar way as previous studies with human ASD subjects. Following the experimental paradigm and model analysis in Lawson et al. (2017), we designed implicit visual learning as a counterpart task for mice, and employed Hierarchical Gaussian Filter (HGF) as an ideal Bayesian observer model to compare neuronal dynamics (or “trajectories”) with model parameter dynamics. We agree that HGF is just one of many possible models for implementing Bayesian inference in the brain. This is, however, the one widely used in the previous human studies (including Lawson et al., 2017), and alternative approaches would make it difficult to interpret our results in this context. We clarified these points in the revised manuscript, and left the questions for future research if and how the brain actually performs Bayesian inference.

2. Model prediction: The visualization of the model predictions in Figure 1D and 1E is confusing. It would be helpful if the authors could present the model predictions in a way that directly relates to the neural activity differences between early stages (ES), late stages (LS), and very late stages (VL) epochs, as shown in later figures. Currently, it is not clear what critical

differences the two autism models make in the LS and VL epochs. Additionally, the text describing the model prediction is also unclear, particularly regarding the dominance of the likelihood in model 1 (line 144; overly high likelihood will dominate posterior and posterior will follow likelihood, not prior). Clarifying this point would improve the paper's clarity.

We apologise for the confusion. We originally hypothesized that anomalies in ASD arise from a failure in establishing either likelihood or prior, which leads to defects in the posterior dynamics during the volatile and stable environments, respectively (**Table R1**):

	likelihood	prior	posterior
WT	high-high-low	low-high-high	low-high-low
ASD model 1 (likelihood too strong)	high-high-high	low-high-high	low-high-high
ASD model 2 (prior too weak)	high-high-low	low-low-low	high-high-low

Table R1: Transition of the distribution precision during implicit learning ("Early Stable - Late Stable - Volatile" epochs). Atypical activity in each ASD model is highlighted with boldface. Note that an opposite polarity is expected for the variance.

However, we cannot directly relate such dynamics to the observed neuronal activity because we do not exactly know how these probability distributions (and the underlying parameters) are represented in the brain. Instead, we took these expected dynamics as a reference to interpret how contextual modulation of the observed neuronal activity may reflect the posterior dynamics (computed somewhere in the brain); and further interpret neurophysiological phenotypes in *Scn2a^{+/-}* mice. To clarify the points and avoid confusions, we revised **Figure 1** and the main text accordingly.

3. Interpreting SC activities in terms of Bayesian models: The paper's interpretation of SC neural activity in relation to the hierarchical Bayesian model is confusing. The authors state that SC encodes learning rate and uncertainty, but this does not align with the activity difference pattern observed between ES, LS, and VL epochs. In the early results section, the authors interpret the low SC activity during LS as indicating lower prediction error, which contradicts the hierarchical Bayesian model comparison. It is important to clearly distinguish between bias (posterior mean - prior/likelihood) and variance (or precision) when linking them to neural data, as they play distinct computational roles in Bayesian models.

We apologise for the lack of clarity that led to the confusion and misunderstanding. We never interpreted the SC activity as indicating a prediction error or explicitly encoding any of the Bayesian model parameters. Instead, we think that the Bayesian inference likely involves higher brain areas, and information about the posterior is transmitted to SC via V1 modulatory input (hence hierarchical as in HGF), leading to the observed contextual modulations of the visual processing in SC.

More specifically, here we analyzed our data in two ways. First, we made population-level global analyses (**Figures 2-6** in the revised manuscript) to qualitatively associate our

observations (summarized in **Table R2**) with the Bayesian inference framework (**Table R1**). In WT mice, modulation patterns of SC cells and V1 input agree with the expected dynamics of the posterior variance; and SC dynamics in Scn2a^{+/-} mice are correlated with the expected dynamics of the posterior variance when the prior is too weak (ASD model 2). Modulation patterns of the pupil dynamics did not align with the Bayesian framework. Nevertheless, contextual modulation was found only in WT mice, and the results of our perturbation experiments support that V1 provides critical signals for modulating SC/pupil dynamics in both WT and Scn2a^{+/-} animals, though V1/pupil dynamics in Scn2a^{+/-} mice did not show significant contextual modulations.

	RGC boutons	SC cells	V1 axons	Pupil size
WT	no modulation	high-low-high	high-low-high	large-large-small
WT with V1 block	N/A	no modulation	N/A	no modulation
Scn2a	no modulation	low-low-high	no modulation	no modulation
Scn2a with V1 block	N/A	no modulation	N/A	no modulation

Table R2: Summary of the observed modulation patterns of the population responses to the aversive stimuli in our experimental paradigm. See Figure 2 for details on SC cells; Figure 5 for SC cells with V1 block; Figure 3 for RGC boutons; Figure 4 for V1 axons; and Figure 6 for pupil dynamics in the revised manuscript. See also Figure 8 for a graphical summary.

We then proceeded with the second analysis for in-depth evaluations at the single-cell level (**Figure 7** in the revised manuscript). In particular, we asked how the observed dynamics of individual neurons are correlated with the parameter dynamics of an ideal observer model (HGF) on a trial-to-trial basis, and found distinct correlation patterns between the brain areas (SC/V1/RGC) and also between the genotypes (WT versus Scn2a^{+/-}). In short,

- 1) much fewer V1 axons were correlated with the model parameter in Scn2a^{+/-} mice than in WT mice, supporting anomaly of V1 input in Scn2a^{+/-} animals.
- 2) A larger fraction of SC cells was correlated with higher-level model parameters in WT mice than in Scn2a^{+/-} mice, or in WT with V1-block. This indicates a critical role of V1 input in SC processing, and anomalous SC processing in Scn2a^{+/-} mice (due to defective V1 inputs).
- 3) In WT mice, a larger fraction of cells was correlated with higher-level model parameters in SC than in V1 axons, suggesting distinct visual processing between the two brain areas.
- 4) Hardly any RGC had significant correlations with the model parameters in both genotypes, suggesting that RGCs have little to do with the learning process.

The presence of correlation in this analysis does not necessarily mean that such information is used for Bayesian computation in the brain. Nevertheless, it offers an interpretable representation of our experimental data, and highlights the difference between SC and V1 dynamics that we could not identify in the first analysis.

Another question that arises is why SC is considered suitable for computing posterior distributions in the Bayesian sense. The fact that most SC neurons did not respond to omitted aversive stimuli is not consistent with the idea that SC is responsible for posterior computation.

Since the horizontal cue still has a probability (17%) of predicting aversive stimuli, SC should generate a response if it were responsible for posterior computation. This discrepancy needs to be addressed and explained.

As detailed above, we never considered that SC was directly involved in the Bayesian inference process itself. Indeed, our data do not support this as the reviewer pointed out.

Furthermore, it is crucial to note that the Bayesian model is applied to the conditional probability $p(\text{outcome}|\text{cue})$, not $p(\text{cue})$ alone. Therefore, the Bayesian inference should be carried out in high-level sensorimotor regions, rather than in the sensory domain. This has implications for the conceptualization and terminology used in the paper. For example, the posterior output does not lead to a "perception" error, as depicted in Figure 1.

We fully agree that, if indeed the brain performs the Bayesian inference, it should involve higher cortical areas, or likely the entire brain. The contextual modulation pattern of SC followed the expected behavior of the posterior, not because SC calculated the posterior, but likely because the information about the posterior was transmitted to SC via the top-down modulatory signals. We clarified the point in the revised manuscript (and left the phrase "perception error" in Figure 1 to describe a general concept behind our experimental design).

4. Neural data analysis: The majority of the analysis in the paper focuses on the period when aversive stimuli are presented. However, both at the Bayesian model and behavioral levels, the cue period also deserves thorough analysis. During the cue period, the prior prediction is likely formed and maintained until the outcome period, where it is integrated with actual outcome information. Analyzing the cue period and its relation to the outcome period would provide valuable insights (Darlington et al., Nat. Neuro., 2019 and Walker et al., Nat. Neuro., 2020).

Following the reviewer's suggestion, we analyzed the response patterns during the two cue stimuli (**Supplementary Figure 2** in the revised manuscript). While adaptation was observed in some cases (e.g., SC responses to the horizontal cue in Scn2a^{+/-} mice, or V1 responses to either cue in WT mice), we generally found no substantial contextual modulation in both neuronal and pupil dynamics (hence no further model analysis was conducted). Therefore, SC is unlikely to be directly involved in the Bayesian inference process in itself, such as the formation of the prior prediction. Note, however, that such response dynamics to the cue stimulus were distinct from those to the aversive stimulus. This highlights the specificity of the contextual modulation in the early visual system. We clarified these points in the revised manuscript.

It would also be beneficial to provide a better characterization of visual responses in SC, as any differences in simple visual responses between WT and Scn2a mice could be critical.

We mapped the receptive field (RF) of the recorded cells by reverse correlation methods, but found no substantial differences between WT and Scn2a^{+/-} mice (except for the RFs of SC cells in WT mice and those in Scn2a^{+/-} mice with V1-block; **Supplementary Figure 6** in the revised manuscript). This indicates that basic response properties of the early visual system remain largely intact in Scn2a^{+/-} mice. We added this new result to the revised manuscript.

Additionally, exploring the effects of different types of aversive stimuli (looming versus sweeping, direction of sweeping) would further enrich the findings.

See our response to the Reviewer #1's comments (item #4). In short, different types of aversive stimuli drove cells in different ways (**Figure R2**); however, we could not further explore the modulation patterns due to low trial numbers and high trial-to-trial variabilities.

5. Minor points:

Distinction of early versus late stable epochs is rather arbitrary. It needs to be justified (as even the 1st block was removed due to movement artifacts). More fine-grained analysis of VL epoch can also reveal something interesting.

We agree that the definition of these analysis "epochs" was rather arbitrary. Following the experimental design in Lawson et al. (2017), here we chose the ES and LS epochs to match the trial number of the VL epoch (72 trials). Due to high trial-to-trial variability, it was difficult to further divide the VL epochs. We instead performed neuronal trajectories analyses (from the HGF model perspective) to better characterize the response dynamics over trials (**Figure 7** in the revised manuscript).

The proportion of neurons correlated with the Bayesian model variables seem too low even though they are statistically significant. Given the total number of neurons is a few hundreds, it must be only a few neurons.

A given neuron can have significant correlations with multiple HGF parameters, and here we assigned such neurons to the parameter with the largest absolute correlation value. Up to tens of neurons (around 5%) were then identified to have significant correlations with a given HGF model parameter. This number adds up across parameters in genotype- and brain area-specific ways so that significant correlations were observed in up to around a third of the recorded cells in total (e.g., 28%, $n=301/1080$ WT SC cells; see **Supplementary Table 2** for details). Thus, a substantial number of neurons were correlated with the HGF model parameters.

Pupil data was not fully explored in that it was not directly analyzed together with neural data. For example, how is it related to Bayesian model variables? Can it be linked to neural data in a trial-by-trial manner?

Following the reviewer's suggestion, we compared the trial-by-trial dynamics of the pupil size and the model parameter (**Figure R3a,b**). We found relatively strong negative correlations across model parameters in WT mice, and these became rather negligible after chemogenetically blocking V1. In contrast, we found relatively weaker negative correlations in *Scn2a^{+/-}* mice, and they largely changed the polarity after blocking V1. Consistent with the analysis results on the pupil size (**Figure 6** in the revised manuscript), this suggests that 1) certain differences exist between WT and *Scn2a^{+/-}* mice at the behavioral level, and 2) the effect of the corticotectal pathway extends to the behavioral level of the responses. We, however, do not have enough samples to test statistical significance.

We also ran a correlation analysis between neuronal and pupil dynamics (**Figure R3c**). In WT mice, we found a significant negative correlation for V1 axons at the population level. In contrast, we found a weaker but significant negative correlation for V1 and RGCs in *Scn2a*^{+/-} mice. This indicates a certain level of miscoordination between the behavioral and neuronal dynamics in *Scn2a*^{+/-} mice. It remains to be addressed, though, if this arises from behavioral defects and/or neuronal defects.

Figure R3: relationship between neuronal and pupil dynamics. **a** Population average of the normalized pupillary trajectories $\langle \Delta a_k^{\text{res}} \rangle_{\text{animals}}$ (gray shade, s.e.m.): from left to right: 21 WT mice, 23 *Scn2a*^{+/-} mice, 5 WT mice with chemogenetic inhibition of V1-to-SC projection, and 6 *Scn2a*^{+/-} mice with chemogenetic inhibition of V1-to-SC projection. **b** Left, Pupillary representation of HGF parameters, given as the average Pearson's correlation between the pupillary trajectory (as in **a**) and the HGF parameters (as in Figure 7b). Right, average correlation per level. From left to right: WT, *Scn2a*^{+/-}, WT with V1-block, and *Scn2a*^{+/-} with V1-block. **c** Distribution of Pearson's correlation between the pupillary trajectories Δa_k^{res} and the neuronal trajectories Δz_k^{res} (open circle, average for each mouse; vertical thick gray bar, interquartile range; horizontal lines, mean \pm s.e.m.). A nested linear mixed-effect modeling was used to assess if the mean is deviated from zero in each case (with Benjamin-Hochberg FDR correction: *, $q < 0.05$; ***, $q < 0.001$).

Reviewer #3 (Remarks to the Author):

Ferrarese and Asari performed calcium imaging, pupillometry, chemogenetics and a hierarchical Bayesian learning model to investigate atypical contextual visual integration in Scn2a^{+/-} mice engaged in an implicit learning task in stable and volatile environments. The results suggest that the superior colliculus (SC), not retinal inputs to SC, drives the observed distinct context-dependent modulation of neuronal and pupillary responses of wild-type and Scn2a^{+/-} mice. Significantly, the authors propose that V1 L5 pyramidal neurons likely mediate contextual visual integration and this corticotectal pathway may underlie abnormal sensory contextual learning in autism spectrum disorder (ASD). There are significant gaps, unfortunately, that make it difficult to evaluate the data fully, largely because data are piecemeal between WT and Scn2a^{+/-} mice, with some major aspects only done in WT mice. A more balanced comparison between the genotypes would strengthen the work considerably.

1) Implicit in the experimental design (and the framing of the work) is the idea that NaV1.2 has a major role in the corticofugal neurons that feed into SC. But the effects of loss of NaV1.2 in other components of the SC integrative pathway are not considered. NaV1.2 appears to be expressed in retinal ganglion cells (PMID 31734278). Is it expressed in SC as well? These components need to be considered before making conclusions that the effects observed are purely mediated by some “top down” effect from V1.

To the best of our knowledge, the expression of Na_v1.2 in the mouse SC was not reported in the literature. However, we cannot deny its expression and the effects on the intrinsic function of the cells. Here we performed a reverse-correlation analysis and characterized the visual response properties of the recorded cells in WT and Scn2a^{+/-} mice. As described above (Reviewer #2, item #4), we found that the RF size of SC cells, RGCs, and V1 axons was comparable between the genotypes (**Supplementary Figure 6** in the revised manuscript). Thus, taken together with the other results, anomalies in Scn2a^{+/-} mice arise likely from defects in V1 and/or higher cortical areas. We clarified the point in the revised manuscript.

2) An understanding of how the pupillometry measurements (and the calcium imaging data) relate to the stimuli is very opaque. Furthermore, consistent experiments across genotypes would be very useful for determining where (and when) deficits arise in Scn2a^{+/-} conditions. The following need to be addressed:

2a) a thorough analysis of these measurements relative to sweeping and looming spots is critical to understand how the genotypes behave in response to stimuli. Do the Scn2a heterozygous mice respond consistently on each trial? Are they as attentive over time? Does the response adapt over time?

In both WT and Scn2a^{+/-} mice, pupil size generally became smaller (constriction) in response to the cue stimuli (e.g., **Supplementary Figure 2** in the revised manuscript), and went back to the baseline during the omitted stimulation (e.g., **Supplementary Figures 3** in the revised manuscript), or became larger (dilation) upon aversive stimulus presentation (e.g., **Figure 6** in the revised manuscript). Such pupil dynamics showed no contextual modulation, except for the dilating responses during the aversive stimulation in WT mice. We cannot make a reliable comparison of such pupil responses to each stimulus type due to a) high trial-to-trial variabilities in both WT and Scn2a^{+/-} mice; and b) low trial numbers per stimulus for each animal. Nevertheless, consistent responses over time across epochs suggest that both WT

and Scn2a^{+/-} mice were attentive throughout the experiments. See also our response to the Reviewer #1's comment #5.

2b) Some evaluation of visual acuity in the Scn2a mice is required, especially because many children with SCN2A loss are diagnosed with cortico-visual impairment. At the very least, experiments described in Fig 3 should also be done in Scn2a^{+/-} mice.

We have performed RGC bouton recordings in WT and Scn2a^{+/-} mice, and found no contextual modulation in our experimental paradigm (**Figure 3** in the revised manuscript). As described above (item #1), the RF size of RGCs, SC cells, and V1 axons was similar between the genotypes (**Supplementary Figure 6** in the revised manuscript). This indicates that the basic functional properties of the early visual system in WT and Scn2a^{+/-} mice are comparable, though there is no direct link with the animals' visual acuity at the behavioral level. These points were clarified in the revised manuscript.

2c) Pupil time series traces should be presented with the accompanying calcium time series traces in each figure. Instead of (or perhaps in addition to) plotting $\langle z(t) \rangle$ for the aversive stimuli of each epoch, a plot of the average neuronal response pre/post presentation of the neutral cue and pre/post aversive cue onset in the same time series should be included. I ask because these data may reveal the anticipatory response of wild-type and Scn2a^{+/-} to the forthcoming non-aversive or aversive stimulus following the neutral cue and glean insight into the overall engagement of the mice with the task.

We have included corresponding pupil dynamics in all the relevant figures in the revised manuscript. We have also included the analysis results on the responses to the cue stimuli (**Supplementary Figure 2**) as well as the omitted stimulus (**Supplementary Figure 3**) in the revised manuscript. Trial-to-trial response examples were also shown for both genotypes in the revised manuscript (**Supplementary Figure 1**).

3) Was there a baseline level activity difference in SC neurons when exposed to the neutral, non-aversive, vertical and aversive stimuli between wild-type and Scn2a^{+/-} mice? In looking at the data from WT mice in Supplementary Fig. 1, SC neuron calcium activity skewed more active when presented with vertical gratings preceding the aversive stimuli. Was this comparable in the Scn2a^{+/-}? Are the receptive fields different? If so, one might expect a bias in task performance based on perception?

As described above in our responses to the reviewer #2's comment #4, we found overall equivalent responses to the two cue stimuli as well as the modulation patterns during the implicit visual learning (**Supplementary Figures 2 and 3** in the revised manuscript). Some SC neurons are orientation selective (Wang et al., 2010; Feinberg and Meister, 2015; de Malmazet et al., 2018), hence their responses are biased to their preferred orientation at the single-cell level. At the population level, however, we found no substantial bias in the neuronal responses.

4) Taking a step back, why was SCN2A chosen amongst all ASD-related genes? ASD is a spectrum, and it is unclear whether children with SCN2A loss of function present with issues as described in Lawson 2017. Have such studies been done within this particular group?

We chose SCN2A for two reasons. First, it is one of the genes most strongly linked to ASD in human studies (Satterstrom et al., 2020). Children with mutations in SCN2A have also demonstrated anomalous acoustic responses at EEG level (Ben-Shalom et al., 2017), suggesting anomalies in sensory perception, though no direct tests on contextual learning (as in Lawson et al., 2017) or visual processing have been reported to our knowledge.

Second, *Scn2a*^{+/-} is one of the best-established ASD mouse models available (Spratt et al., 2019). It encodes Na_v1.2 subunit that has a clear relevance to neuronal function. Furthermore, it has been reported that *Scn2a*^{+/-} mice have defects in Layer 5 corticofugal neurons (Spratt et al., 2019). This made it a top choice among many other model animals, as we aimed to characterize how SC cells integrate bottom-up retinal inputs and top-down V1 inputs.

5) The control for DREADDs is incorrect. One needs a sham virus with CNO, not saline, given the well-known off target effects of CNO across the brain.

We have performed proper control experiments to compare the responses in WT mice with/without hM4D expression with/without CNO injections, and confirmed that CNO and hM4D had no effect in itself (**Supplementary Figure 5** in the revised manuscript). See also our response to the Reviewer #1's last item on the same issue.

6) It is unclear how statistics were done with large populations of cells within a relatively smaller population of mice. There will be inter-animal variability; was this taken into consideration? I.e., were the data analyzed in a nested fashion? It was unclear whether this was the case from the text provided.

Yes, we have used nested ANOVA. We revised the methods section for clarification.

Minor:

Fig 4b does not show that injections in V1 reach SC. A supplemental image showing mCherry in SC is important to show. Furthermore, it is critical to know how large of a population of V1 cells is labeled across genotypes to interpret properly the hM4Di results.

We have included a proper example in the revised manuscript (**Figure 5b**), demonstrating mCherry signals (i.e., V1 axons) in SC. On average, about a third of the V1 L5 areas were labelled. See also our response to the reviewer #1, comment #2.

Methods: "Analysis of neuronal activity contextual modulation" - How was the threshold value determined for selecting responsive cells for subsequent analyses? Based on this thresholding, a range of 22 to 99 cells does not seem balanced, or I may misunderstand.

In the original manuscript, we set an arbitrary threshold for each recording to focus only on "well-responsive" cells to minimize the effects of noise (trial-to-trial variability). In the revised manuscript, we used a fixed threshold (z-score larger than 1 for somata and 0.75 for boutons) for all recordings. While the trial-to-trial variability became somewhat higher, the overall conclusion remained the same. Thus, we believe the robustness of the reported results.

Line 126 "we presented aversive outcome stimuli more frequently after one neutral cue stimulus than after another." Rephrase for clarity, please.

We apologize for the confusion, and clarified the text in the revised manuscript. Here we meant that the aversive stimulus was preceded by the vertical cue stimulus more frequently (83%) than the horizontal cue stimulus (17%).

We thank the reviewers for their continued support and constructive comments (in blue). Below are our point-by-point replies (in black).

Reviewer #1 (Remarks to the Author):

The authors have addressed my previous concerns with additional experiments and analysis, providing convincing responses. Most of my initial concerns stemmed from misunderstanding aspects of the experimental protocol and analysis. However, I remain unclear on two key points that would benefit from clarification in the methods section:

Silencing coverage: Why was optogenetic silencing applied to such a small portion of the visual field (median 22% of V1)? Does the author look exactly at the same cells in SC? This limited coverage may lead the authors to underestimate their results.

Only a small population of V1 neurons was labelled because, using viral tools, we selectively targeted only those V1 cells projecting to the SC area where recordings were made. Specifically, here we injected:

- Cre-dependent AAVs carrying hM4Di into V1, and
- retrograde AAVs into SC to deliver Cre recombinase to the cells projecting to SC.

To monitor the activity of local SC cells at the injection site, we co-injected AAVs carrying GCaMP along with the retrograde AAVs. The recorded SC cells were thus surrounded by the silenced V1 axons, thereby maximizing the chance to observe the effects of the chemogenetic manipulation. However, experiments with and without CNO injection were conducted in different sessions. Therefore, although we made every effort to return to the same field of view under the microscope, we cannot guarantee that the results were based on recordings from the exact same SC cells.

Receptive field analysis: Regarding my previous major comment 3 about potentially underestimating the number of modulated neurons - if the authors mapped neuronal receptive fields, shouldn't the analysis only include neuronal responses when visual stimulation falls within each neuron's receptive field? This methodological detail needs clarification.

It is not readily feasible to determine exactly when the visual stimulus reached the receptive field of each neuron due to an animal's eye movements. Furthermore, the activity of SC neurons can be modulated by stimuli outside the classical receptive field. For these reasons, we based our analysis on the entire stimulation period.

Given that several of my original concerns arose from protocol ambiguities, I recommend the authors provide more detailed methodological descriptions to prevent similar misunderstandings for future readers.

We included further details of the above points in the revised manuscript.

Reviewer #2 (Remarks to the Author):

The authors have made a commendable effort to address the reviewers' feedback, with new experiments enhancing the specificity of the V1-to-SC projection's changes in different task environments.

However, I have reservations regarding the use of the Bayesian framework in the paper. While the approach itself is not incorrect, the Bayesian model may detract from the paper's strengths, which lie in the detailed circuit dissection and well-designed behavioral tasks. Specifically, I recommend removing Figure 1c and related text from the Results section for several reasons. Firstly, Figure 1c is only conceptual and does not directly generate predictions on the neural data ('high-low-high' activity across 'ES-LS-VL' epochs), leading to ambiguity in its interpretation. Questions arise, such as the representation of likelihood and prior - are they related to aversiveness, perception error, or cue probability? The hierarchical Bayesian model in later figures provides a more detailed and precise understanding of multiple neural signals. Therefore, removing Figure 1c would not diminish the paper's quality.

Secondly, Figure 1c and its text do not address bias or prediction error, which are crucial first-order statistics when considering variance, a second-order statistic. The Bayesian model's core idea involves a bias-variance trade-off, and focusing solely on variance is incomplete.

Thirdly, the text accompanying Figure 1c appears to offer post-hoc explanations to fit the data, rather than presenting a quantitative model. Questions arise, such as why the volatile environment "lowers likelihood" or why likelihood and prior are assumed to be Gaussian distributions. Did animals have more sensory noise for aversive or non-aversive stimulus? And why does "the prior belief remained relatively high and precise because updating the prior is slow in general"? Didn't animals learn that cue-outcome contingency from late stable epoch is not valid anymore in a volatile environment?

An alternative to Figure 1c could be to generate predictions on 'outcome uncertainty'. In the early stable environment, the outcome distribution is bimodal, with equal aversive and non-aversive outcomes. Through learning in the stable epoch, this distribution becomes skewed, leading to reduced outcome uncertainty (similar to posterior variance). In the volatile environment, WT animals would learn a less skewed bimodal distribution, resembling the early stable epoch. This approach does not require the Bayesian framework and offers a more parsimonious explanation. The authors could still fully utilize the Bayesian model later with Figure 7.

Ultimately, the decision on Figure 1c rests with the authors. Congratulations on the work.

We are deeply grateful to the reviewer for such thoughtful and thorough comments on our work. We decided to remove Fig. 1c and relevant texts in the revised manuscript.

Reviewer #3 (Remarks to the Author):

The authors have performed additional experiments and analysis that improve the manuscript considerably. There is one aspect that should be amended. In response to concern #3: Was there a baseline level activity difference in SC neurons when exposed to the neutral, non-aversive, vertical and aversive stimuli between wild-type and Scn2a^{+/-} mice? The authors should make direct comparisons demonstrating that responses to horizontal and vertical cues differ between WT and Scn2a^{+/-} rather than relying on population averages. These are critical values that are needed to better interpret the calcium responses to aversive stimuli.

We appear to have misunderstood the reviewer's point before. To address the question, we reanalyzed our data, and found no substantial differences between WT and Scn2a^{+/-} mice in the response bias of SC cells between horizontal and vertical stimuli.

Specifically, by taking the responses during the early stable period as the baseline, we first examined the stimulus preference at the single-neuron level. As expected from the diverse tuning properties of SC cells (Wang et al., 2010, J Neurosci), we found that some cells responded preferentially to vertical stimuli, while others to horizontal ones. In both WT and Scn2a^{+/-} mice, about 20% of the recorded SC cells showed such orientation-selective responses (WT, 23±10%; Scn2a^{+/-}, 22±9%), with an overall response preference to vertical orientation (**Table R1**).

Table R1: The response patterns of SC cells to the two cue stimuli in WT (left) and Scn2a^{+/-} mice (right). See also **Supplementary Table 1**.

WT (15 animals)		Response to vertical gratings	
		Yes	No
Response to horizontal gratings	Yes	N=287	N=162
	No	N=285	N=1251

Scn2a ^{+/-} (15 animals)		Response to vertical gratings	
		Yes	No
Response to horizontal gratings	Yes	N=209	N=215
	No	N=255	N=1444

We then examined the response preference in each animal, and found that data from some animals showed a bias toward vertical stimuli (8 WT and 6 Scn2a^{+/-} mice, $p < 0.05$ by chi-square test with Bonferroni correction), while those from other animals toward horizontal stimuli (4 WT and 2 Scn2a^{+/-} mice). This batch effect arose because the mouse SC has patchy orientation maps (Feinberg and Meister, 2015, Nature; Li et al 2020, Curr Biol), where neighboring SC cells tend to have a similar orientation selectivity. Importantly, the medial-posterior part of the SC we monitored is known to have an overall bias toward vertical orientation (in the dorso-temporal visual field; we updated the schematic in **Figure 2a** and **Supplementary Figure 6** for clarification), whereas other parts of SC toward horizontal orientation (Ahmadlou and Heimel, 2015, Nat Commun; de Malmazet et al., 2018, Curr Biol). Thus, the presence of the response bias in the recorded cells does not necessarily mean that the entire SC population has the corresponding bias, not to mention the animal's perception. Overall, our animal-by-animal analysis revealed no significant difference in the population response bias between WT and Scn2a^{+/-} mice ($p = 0.31$, Kolmogorov-Smirnov test; $N = 15$ each; **Figure R1**). We clarified these points in the revised manuscript.

Figure R1: Response preference of the superior colliculus neurons in WT and *Scn2a*^{+/-} mice between vertical over horizontal stimuli.

a,b: We first selected those cells that were responsive to at least one cue stimulus, $\max[z_v(t)] > 1$ and/or $\max[z_h(t)] > 1$, where $z_v(t)$ and $z_h(t)$ are the trial-average responses to the vertical and horizontal gratings, respectively. We then calculated the difference of these responses, $z_v - z_h = \langle z_v(t) \rangle_t - \langle z_h(t) \rangle_t$, for each cell, and showed the distribution for each animal (**a**, WT mice; **b**, *Scn2a*^{+/-} mice); circles, median; whiskers, 25 and 75 percentiles. Data from some animals were significantly biased: **, $p < 0.01$; *, $p < 0.05$, Wilcoxon signed-rank test with Bonferroni correction.

c: While certain response bias was observed on an animal-by-animal basis, there was no overall difference between WT and *Scn2a*^{+/-} mice ($p = 0.31$, Kolmogorov-Smirnov test, based on median[$z_v - z_h$] per animal; $N = 15$ each).

We thank the reviewers for their continued support and constructive comments (in blue). Below are our point-by-point replies (in black).

Reviewer #1 (Remarks to the Author):

Thank you for your responses. Regarding my two previous comments:

1- I would suggest clearly stating the coverage and providing a thorough discussion of this aspect.

The revised manuscript includes details on the silencing coverage. Specifically, our histological analysis confirmed that axonal projections were visible in the ipsilateral SC of all mice examined (e.g., **Fig. 5b**; n=7 each for WT and Scn2a^{+/-} mice); and these projections arose mostly from the V1 L5 area, where labelled cells occupied 31±22% (mean ± standard deviation), mostly in the medial-anterior part. Here we aimed at silencing only those V1 neurons that project to SC, but not the entire V1 cells. This is critical not to interfere with the cortical computation, but only inhibit the corticotectal pathway. Previous studies have shown that V1 L5 pyramidal neurons consist of (at least) three subpopulations, and only one of them projects to SC (Kim et al., 2015, Neuron). Thus, the silencing coverage achieved with our viral approach matched expectations.

We also clarified that the experiments with and without CNO injection were conducted in different sessions; hence, the recorded SC populations were not necessarily the same even from the same animal. Nevertheless, we have a high expectation to observe the effects of the chemogenetic manipulation because these recorded SC cells were well surrounded by the silenced V1 axons via co-injection of AAVs carrying GCaMP into SC together with the retrograde AAVs to target V1 axons in SC.

2- If mapping was performed, you should be able to determine when the stimulus enters the receptive field (RF). The surround RF activation is likely minimal using this protocol, and spontaneous eye movements in well-habituated/trained subjects are also minimal. Numerous studies conducting mapping have demonstrated that it does not significantly affect the response for a given firing rate (on average). If you determine that you cannot perform this analysis, you should at minimum discuss this limitation and/or clearly state your experimental conditions.

Many SC neurons showed a strong onset response to a looming stimulus, even though the stimulus did not fall in their receptive field (e.g., **Fig. 2d**; see also Lee et al., 2020, eLife). This suggests that substantial circuit computations beyond the classical receptive field underlie the observed SC responses. Furthermore, the low number of repetitions for each aversive stimulus made it difficult to reliably assess neuronal responses at the single-cell level, especially due to eye movements. We thus decided to stay focused on the average population response during the stimulus presentation in this study. We clarified this limitation in the revised manuscript.

Reviewer #2 (Remarks to the Author):

The modifications the authors made address my comments.

Thanks again for the constructive comments.

Reviewer #3 (Remarks to the Author):

Thank you for the clarification and for providing updated information regarding baseline responses in SC. I have no further concerns, and congratulate all the authors on an impressive work.

Thanks again for the constructive comments.